# Neural Entropy

**Akhil Premkumar**
Department of Applied Physics
Yale University
New Haven, CT 06511, USA
`akhil.premkumar@yale.edu`
*Work done while at the University of Chicago*

## Abstract

We explore the connection between deep learning and information theory through the paradigm of diffusion models. A diffusion model converts noise into structured data by reinstating, imperfectly, information that is erased when data was diffused to noise. This information is stored in a neural network during training. We quantify this information by introducing a measure called *neural entropy*, which is related to the total entropy produced by diffusion. Neural entropy is a function of not just the data distribution, but also the diffusive process itself. Measurements of neural entropy on a few simple image diffusion models reveal that they are extremely efficient at compressing large ensembles of structured data.

## 1 Introduction

How much information is stored in a neural network? As a simple example, consider training a neural network to store an 8-bit grayscale image of dimension $H \times W$ pixels. The network learns a smooth map from pixel co-ordinates to grayscale intensity values from $H \times W$ bytes of raw data. This is not the total number of bytes of the parameters that constitute the network, and not every image of size $H \times W$ contains the same amount of information. But it is reasonable to expect that if we push images of higher and higher resolutions/detail onto the same network, at some point the network will not be able to reproduce the images faithfully

The question is even more pertinent in the context of generative models. These models are capable of producing seemingly endless variations of the original training data, say images, but that does not mean the neural network has stored an infinite number of images. Rather, generative models store a *distribution* of images, call it $p_d$, and the generated samples are points that interpolate the training data in $p_d$. This is similar to how the network from the prior example blends the grayscale intensities between neighboring pixels. So the analogous question to ask is this: how many bytes of data is $p_d$ worth? The primary goal of this paper is to answer this question in the context of diffusion-based generative models (hint: it is not simply the Shannon entropy of $p_d$, see App. C.2).

Diffusion models serve as a natural bridge between information theory and machine learning, having been inspired by ideas from non-equilibrium thermodynamics [1], which itself can be viewed as an application of information-theoretic principles to physical systems [2–4]. Very briefly, samples from a training dataset are incrementally noised till they are distributed as a generic Gaussian, call it $p_{eq}$, while a neural network learns to reverse these noising steps. Once trained, the network can transform a random Gaussian vector into a highly structured output that resembles a typical member of the training data. In the continuum limit, the noising and denoising stages become diffusive processes [5, 6], the thermodynamic properties of which are well established [7–9].

Diffusion gradually wipes out information from $p_d$ over time (cf. Fig. 6). The information loss is quantified by the total entropy produced during the process, $S_{tot}$. Within this framework, we can

define the information content of $p_{\text{d}}$ in relation to the diffusion process itself—it is the amount of information that must be reinstated to drive the process away from its equilibrium state $p_{\text{eq}}$ back to $p_{\text{d}}$. It is precisely $S_{\text{tot}}$ (cf. App. C). A well-trained diffusion model retains nearly all of this information in its neural network. Therefore, we can characterize the information content of the network by a quantity we call the *neural entropy*, $S_{\text{NN}} \approx S_{\text{tot}}$, defined in Eq. (18).

Before we delve into the details, a few points must be clarified. First, it is important to stress that neural entropy quantifies the information stored in a perfectly trained network; it is *not* the entropy of the phase space density over the neural network's internal microstates. Second, no diffusion model can reconstruct $p_{\text{d}}$ perfectly because we only have access to a finite number of training samples from it [10], and training is imperfect even with a large dataset. Third, the neural network encodes and interpolates the given information, drawing from its own inductive biases to fill in the gaps between the training data [11]. This is why diffusion models are able to estimate very high-dimensional distributions even from relatively small datasets [12]. Consequently, neural entropy is just one part of a slew of variables, like the choice of network architecture, optimization algorithm, etc. that ultimately affect the overall model performance.

Despite these caveats, empirical measurements of neural entropy reveal interesting insights into the behavior of neural networks. First, in a setting where the inductive biases are relatively weak and the data distribution is largely unstructured (e.g. Gaussian mixtures), diffusion models tend to struggle to reconstitute $p_{\text{d}}$ accurately as more information is fed into the network (see Fig. 10). Second, in image diffusion models with U-nets trained on real images, the neural entropy shows a distinct *logarithmic* scaling with the number of training samples $N$ (see Figs. 1 and 16). That is, the marginal information gained per sample decreases approximately as $1/N$. The quality of generated images also reflects this trend (see Fig. 18). Provided that $N$ is sufficiently large for the model to approximate $p_{\text{d}}$ well, diffusion models compress the images with great efficiency, since they encode the ensemble statistics of the training data; storing each image separately in old-fashioned memory would have incurred a linear cost in $N$.

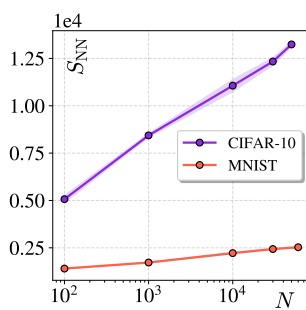

Figure 1: Neural entropy vs. number of samples for two image diffusion models.

## 2   Schrödinger's Gedankenexperiment

The link between diffusion and information theory can be traced back to a thought experiment introduced by Erwin Schrödinger in a seminal paper from 1931 [13]. Consider a diffusion process like the dissolution of an ink drop in water. Common experience suggests that the ink particles would homogenize over the available volume of water and remain in this diffused state indefinitely. However, there is a very small but non-zero probability that the ink particles collect together in some exotic configuration at a later time. Schrödinger asks: what is the probability that the particles diffuse back to their original state?

To answer this question in a simpler setting we study random walkers on a one-dimensional lattice. The lattice sites are spaced by $\ell$ and the walkers jump to one of their nearest neighbors at each time step. The density of walkers at $x$ updates as

$$p(x, t + \Delta t) = q_R(x - \ell)p(x - \ell, t) + q_L(x + \ell)p(x + \ell, t), \tag{1}$$

where $q_R(x)$ (or $q_L(x)$) is the probability that a walker at $x$ jumps to the right (or left) at time $\Delta t$ (see App. B.1 for notation). But that does not mean exactly $q_R(x)$ fraction of all walkers at $x$ always jump rightwards in $\Delta t$; over several trials, there will be small fluctuations in the actual number of walkers that make such a transition. Such fluctuations can accumulate to evolve $p$ in a manner different from Eq. (1), albeit with low probability. One may liken this to throwing a perfectly fair coin 1000 times: due to fluctuations, we do not always obtain the expected outcome of 500 heads and 500 tails, and in fact there is a minute probability of $2^{-1000}$ of obtaining all heads (or all tails).

For appropriate choices of $q_R(x)$ and $q_L(x)$, there exists an equilibrium distribution $p_{\text{eq}}$ which satisfies the detailed balance conditions corresponding to Eq. (1). Let $T$ be a large enough time that

the walkers can equilibrate to very nearly $p_{eq}$ from another state $p_d$. Starting from $p_{eq}$ at $t = 0$, the probability that the walkers would migrate back to the distribution $p_d$ at time $t = T$ is [13, 14]

$$\mathcal{P}[p_d] \propto \exp\left[-M \sum_{x_0, x_T} p_{eq}(x_0) h(x_T|x_0) \log \frac{h(x_T|x_0)}{g(x_T|x_0)}\right] \equiv e^{-M D_{KL}(h\|g)}. \tag{2}$$

Here, $M$ is the total number of walkers on the lattice and $g(x_T|x_0)$ is the probability that a walker at $x_0$ ends up at $x_T$ under Eq. (1). That is, $g$ is the transition kernel for Eq. (1). On the other hand $h(x_T|x_0)$ is a kernel that transports $p_{eq}(x_0)$ to $p_d(x_T)$. There are many kernels $h$ that accomplish this, but for sufficiently large $M$ the exponential in Eq. (2) picks out an optimal kernel $h_\star$ for which the Kullback-Leibler divergence $D_{KL}(h_\star\|g)$ is minimum. It can be shown that the evolution of $p_{eq} \to p_d$ under $h_\star$ is a reversal (playback) of the transformation $p_d \to p_{eq}$ under Eq. (1) [14, 15].

With $p_{eq}$ (or $g$) fixed, $\mathcal{P}$ can be understood as a *distribution of distributions*. A sample from $\mathcal{P}$ is a distribution that $p_{eq}$ can fluctuate into at time $t = T$, under the dynamics in Eq. (1). If $M$ is large, $\mathcal{P}$ is sharply peaked at $p_{eq}$; the probability that the walkers would deviate from this configuration is exponentially small. This is true even with $h_\star$—natural processes have a preferred direction of time, and they rarely evolve in reverse. Eq. (2) is intimately related to the Second Law of Thermodynamics. In fact,

$$S_{tot} := \sum_{t=0}^{T-\Delta t} \sum_{x_{t+\Delta t}, x_t} p(x_t, t) h_\star(x_{t+\Delta t}|x_t) \log \frac{h_\star(x_{t+\Delta t}|x_t)}{g(x_{t+\Delta t}|x_t)} \geq D_{KL}(h_\star\|g). \tag{3}$$

where $p(x_t, t)$ is the distribution of walkers as they evolve between $p_{eq}$ and $p_d$, and $S_{tot}$ is the total entropy generated if $p_d$ was subjected to Eq. (1) for time $T$ (cf. App. A.2). In simple terms, $S_{tot}$ quantifies the time irreversibility of the process $p_d \to p_{eq}$ [9, 16, 17]. We discuss the meaning of $S_{tot}$ in greater detail in the upcoming sections and App. C.

Combining Eqs. (2) and (3), we obtain a key relation between the Shannon information content of the outcome $p_d$, per walker, and the total entropy [2, 18, 19]:

$$\frac{S_{tot}}{\log 2} \geq -\frac{1}{M} \log_2 \mathcal{P}[p_d]. \tag{4}$$

If we observe a set of random walkers that was initially at equilibrium and find that they are still distributed as $p_{eq}$ we learn nothing new; that was the outcome we expected. However, in the unlikely event that we observe the random walkers distributed as $p_d$, we would gain an amount of information commensurate with the total entropy generated in diffusing $p_d \to p_{eq}$.

## 3   Diffusion models and Maxwell's demon

We can enhance the probability of obtaining the outcome $p_d$ by bringing $g$ closer to $h_\star$. That is, we adjust the jump probabilities in Eq. (1) such that the distribution of walkers evolves to $p_d$ after time $T$. The modified dynamics reshapes the distribution $\mathcal{P}$ to be peaked around $p_d$ rather than $p_{eq}$

To see how this is implemented in a diffusion model we convert the discrete random walker setup from Eq. (1) to a continuous diffusion process by making the lattice spacing $\ell$ small. Taylor expanding in $\ell$ and keeping the leading terms, we obtain the Fokker-Planck equation (see App. A)

$$\partial_t p(x, t) = -\partial_x (b_+(x) p(x, t)) + \frac{\sigma^2}{2} \partial_x^2 p(x, t), \tag{5}$$

$$b_+(x) := \frac{\ell}{\Delta t}(q_R(x) - q_L(x)), \quad \sigma^2 := \frac{\ell^2}{\Delta t}. \tag{6}$$

We restrict ourselves to drift terms $b_+(x)$ that are confining so that $p_{eq}(x) \propto \exp(\int^x 2b_+/\sigma^2)$ exists. By explicit calculation of Eq. (3), it can be shown that if a distribution $p_d$ is subjected to Eq. (5) the total entropy produced after time $T$ is (cf. Eq. (39) and [9])

$$S_{tot} = \int_0^T dt \frac{\sigma^2}{2} \mathbb{E}_{p(\cdot, t)}\left[\|\partial_x \log p_{eq} - \partial_x \log p\|^2\right], \tag{7}$$

where the expectation value is taken over the distribution $p$ that interpolates $p_{\mathrm{d}}$ and $p_{\mathrm{eq}}$. The r.h.s. in Eq. (7) is the KL divergence between path measures of two stochastic differential equations (SDEs),

$$\mathrm{d}X_t = -(b_+(X_t) - \sigma^2 \partial_x \log p(X_t, t))\mathrm{d}t + \sigma \mathrm{d}B_t, \tag{8a}$$

$$\mathrm{d}X_t = b_+(X_t)\mathrm{d}t + \sigma \mathrm{d}B_t, \tag{8b}$$

upto a boundary term that vanishes when $T$ is large [20]. Eqs. (8a) and (8b) that correspond to the transition kernels $h_\star$ and $g$ respectively (cf. App. A.1). That is, if we reset the clock to $t = 0$ and apply Eq. (8a) for a time $T$ we can drive $p_{\mathrm{eq}}$ back to $p_{\mathrm{d}}$ along $p$. Bringing Eq. (8b) closer to Eq. (8a) would concentrate $\mathcal{P}$ around $p_{\mathrm{d}}$. In a diffusion model this can be done by changing Eq. (8b) to

$$\mathrm{d}X_t = (b_+(X_t) + \sigma^2 \boldsymbol{e_\theta}(X_t, t))\mathrm{d}t + \sigma \mathrm{d}B_t, \tag{9}$$

where $\boldsymbol{e_\theta}(X_t, t)$ is the output of a neural network trained to minimize an equivalent of (cf. Eq. (90))

$$\mathcal{L}_{\mathrm{EM}} := \int_0^T \mathrm{d}t \frac{\sigma^2}{2} \mathbb{E}_p \left[ \|\partial_x \log p_{\mathrm{eq}} - \partial_x \log p + \boldsymbol{e_\theta}\|^2 \right]. \tag{10}$$

It follows from Eq. (4) that, a perfectly trained network stores *at least* the same amount of information as we would learn from observing $p_{\mathrm{eq}}$ fluctuate to $p_{\mathrm{d}}$ under Eq. (8b). In practice, training is not perfect, so the information absorbed by the network is not exactly $S_{\mathrm{tot}}$, as we discuss below. We define the *ideal* neural entropy as the information retained by the network under perfect training,

$$\hat{S}_{\mathrm{NN}} := S_{\mathrm{tot}}. \tag{11}$$

This discussion is reminiscent of Maxwell's demon, a famous thought experiment from physics [21, 22]. The crucial difference is that the demon does not perform work on the system; it measures the state of the system to make decisions about closing doors or adjusting potentials [23]. Diffusion models *do* expend work to reconstitute $p_{\mathrm{d}}$ from $p_{\mathrm{eq}}$, through the modified drift term in Eq. (9). The additional $\sigma^2 \boldsymbol{e_\theta}$ term reshapes the *free energy* landscape to make $p_{\mathrm{d}}$ the most probable outcome (cf. App. C.2). But these models also measure and store state information from simulations of $p_{\mathrm{d}} \to p_{\mathrm{eq}}$ during training.

A true Maxwell's demon would reverse diffusion by waiting for $p_{\mathrm{eq}}$ to fluctuate into $p_{\mathrm{d}}$, an event it learns about by measurement, and switch up the potential to lock $p_{\mathrm{d}}$ into place. This is an example of an 'information ratchet' [24, 25]. On the other hand, a diffusion model remembers $p_{\mathrm{d}}$ in a manner closer to how we store, say, an image in memory. A grayscale image of dimensions $H \times W$ is a sample from a uniform probability distribution over the hypercube $[0, 255]^{H \times W}$. The information gained from observing any sample is $\log_2(256)^{H \times W} = H \times W$ bytes. This is also the amount of information we need to specify to locate a specific sample/image in the hypercube. In the same way, $S_{\mathrm{tot}}$ is the information required to locate within the paths generated by Eq. (8b) a set of paths that transport $p_{\mathrm{eq}} \to p_{\mathrm{d}}$.

## 4 Entropy matching

Having introduced the total entropy $S_{\mathrm{tot}}$ in the context of random walkers on a lattice, we can generalize it to a $D$-dimensional continuous diffusion process with little effort. We will make the drift and diffusion diffusion coefficients time-dependent, but keep the latter isotropic. That is, $p_{\mathrm{d}}$ diffuses under

$$\mathrm{d}Y_s = b_+(Y_s, s)\mathrm{d}s + \sigma(s)\mathrm{d}\hat{B}_s, \tag{12}$$

where we have introduced a new time variable $s := T - t$ for the 'forward' evolution (see Fig. 6). Let $p_0$ be the result of evolving $p_{\mathrm{d}}$ for a time $T$ with Eq. (12). The SDEs from Eq. (8) are updated to

$$\mathrm{d}X_t = -(b_+(X_t, T - t) - \sigma(T - t)^2 \nabla \log p(X_t, t))\mathrm{d}t + \sigma(T - t)\mathrm{d}B_t, \tag{13a}$$

$$\mathrm{d}X_t = b_+(X_t, T - t)\mathrm{d}t + \sigma(T - t)\mathrm{d}B_t, \tag{13b}$$

There is no longer a static equilibrium state since Eq. (13b) changes over time; if we start with $p_0$ at time $t = 0$ and evolve under Eq. (13b) we will obtain a distribution different from $p_0$, which we denote as $p_{b_+}$, that depends on $p_0$ as well as Eq. (13b). But it is useful to define the *quasi-invariant* distribution, $p_{\mathrm{eq}}^{(t)}(x)$, which satisfies the homogeneous Fokker-Planck equation

$$0 = -\nabla \cdot (b_+(x, T - t)p_{\mathrm{eq}}^{(t)}) + \frac{1}{2}\sigma(T - t)^2 \nabla^2 p_{\mathrm{eq}}^{(t)} \implies p_{\mathrm{eq}}^{(t)}(x) = \frac{1}{Z_t} \exp\left[ \int^x \mathrm{d}\bar{x} \frac{2b_+}{\sigma^2} \right]. \tag{14}$$

Intuitively, $p_{\mathrm{eq}}^{(t)}$ can be understood as the 'least informative state' at time $t$. It is the distribution that would result if we froze $b_+$ and $\sigma$ at their values at $t$ and waited for the system to equilibrate. Therefore $p_{\mathrm{eq}}^{(t)}$ depends only on the drift and diffusion coefficients at $t$ and has no memory of the initial state. For example, if $b_+ = -(x - t)$ and $\sigma = 1$ the quasi-invariant state would be $p_{\mathrm{eq}}^{(t)} \propto \exp(-(x - t)^2)$ [7]. Thus, $p_{\mathrm{eq}}^{(t)}$ is the natural generalization of $p_{\mathrm{eq}}$ for time-dependent dynamics. In this paper, we restrict ourselves to forward processes for which the drift and diffusion coefficients have the same time-dependence, that is, $b_+/\sigma^2 = \mathrm{const}$. In that case, it can be shown that

$$ S_{\mathrm{tot}} \equiv \int_0^T \mathrm{d}t \, \frac{\sigma^2}{2} \mathbb{E}_p \left[ \left\| \nabla \log p_{\mathrm{eq}}^{(t)} - \nabla \log p \right\|^2 \right] = \mathrm{D}_{KL} \left( p_{\mathrm{d}} \big\| p_{\mathrm{eq}}^{(T)} \right) - \mathrm{D}_{KL} \left( p_0 \big\| p_{\mathrm{eq}}^{(0)} \right) . \quad (15) $$

This relation is derived in App. B.3. It bears a strong likeness to an important result in thermodynamics called the *Jarzynski equality* [26], specifically the form given in [27]. According to this relation, total entropy is the information gap between $p$ and the maximally ignorant state at a given $T$. Further discussion of the connection to thermodynamics is given in App. C.1. Our main goal is to understand the consequences of Eq. (15) to diffusion models.

Replacing $p_{\mathrm{eq}}^{(t)}$ in Eq. (15) with $p_{b_+}$ turns it into an inequality,

$$ S_{\mathrm{tot}} \equiv \int_0^T \mathrm{d}t \, \frac{\sigma^2}{2} \mathbb{E}_p \left[ \left\| \nabla \log p_{\mathrm{eq}}^{(t)} - \nabla \log p \right\|^2 \right] \geq \mathrm{D}_{KL} \left( p_{\mathrm{d}} \big\| p_{b_+} \right) \quad (16) $$

This is a slight variation of Theorem 1 from [20]. A detailed proof is given in App. B.2. If we modify the drift term in Eq. (13b) to $b_+ + \sigma^2 e_{\theta}$ as we did in Eq. (9), Eq. (16) changes to (cf. Eq. (47))

$$ \int_0^T \mathrm{d}t \, \frac{\sigma^2}{2} \mathbb{E}_p \left[ \left\| \nabla \log p_{\mathrm{eq}}^{(t)} - \nabla \log p + e_{\theta} \right\|^2 \right] \geq \mathrm{D}_{KL} \left( p_{\mathrm{d}}(\cdot) \| p_{\theta}(\cdot, T) \right) . \quad (17) $$

The l.h.s. is the training objective, $\mathcal{L}_{\mathrm{EM}}$, the minimization of which can now be seen as tightening the KL divergence between true $p_{\mathrm{d}}$ and the reconstructed distribution $p_{\theta}$. We call this the *entropy-matching* objective. It is nearly the same as the flow-matching objective from [28], except for the factors multiplying the expectation value.

The neural entropy defined in Eq. (11) is not always the true measure of the information stored in the network. This is often beneficial; if the neural network stored $S_{\mathrm{tot}}$ perfectly for a relatively sparse dataset, such as images, the diffusion model would learn to reconstruct a series of Dirac delta functions in pixel space. We propose that the actual value of neural entropy is estimated by

$$ \boxed{ S_{\mathrm{NN}} := \int_0^T \mathrm{d}s \, \frac{\sigma^2}{2} \mathbb{E}_p \left[ \| e_{\theta} \|^2 \right] . } \quad (18) $$

The time integral has been expressed in terms of $s$ here because entropy is produced in the $s$-direction. Practically Eq. (18) is computed by simulating the forward process Eq. (12) and taking the Monte Carlo average (cf. Eq. (23)). So the expectation is still taken with respect to the ideal reverse evolution.

A relation analogous to Eq. (17) can be derived for score-matching diffusion models by switching the drift term in Eq. (13b) to $-b_+ - \sigma^2 s_{\theta}$ (cf. Eq. (46)),

$$ \int_0^T \mathrm{d}t \, \frac{\sigma^2}{2} \mathbb{E}_p \left[ \| s_{\theta} - \nabla \log p \|^2 \right] \geq \mathrm{D}_{KL} \left( p_{\mathrm{d}}(\cdot) \| p_{\theta}(\cdot, T) \right) . \quad (19) $$

However, setting $s_{\theta} = 0$ on the l.h.s. does *not* give us a term that can be interpreted sensibly as an entropy. For example, if we consider the special case where $b_+$, $\sigma$ are time-independent and choose $p_{\mathrm{d}} = p_{\mathrm{eq}}$, we see that $S_{\mathrm{tot}}$ vanishes and no information would be stored in the neural network in an entropy-matching model. However, $\mathbb{E}[\| \nabla \log p \|^2] \neq 0$ since the score function is non-zero over the support of $p_{\mathrm{eq}}$, so the network ends up having to store 'information' to convert $p_{\mathrm{eq}}$ to itself! Comparing Eqs. (17) and (19) we see that setting $s_{\theta} = \nabla \log p_{\mathrm{eq}}^{(t)} + e_{\theta}$ makes both approaches equivalent, in principle. However, the score-matching network must put additional effort into learning the quasi-invariant distribution, which complicates the interpretation of score-matching loss as an entropy. See App. D for further discussion.

# 5 Thermodynamic uncertainty

Returning for a moment to the random walkers on a discrete lattice, it is apparent that the walkers are less likely to fluctuate into a $p_\text{d}$ that is far different from $p_\text{eq}$, compared to one that is more similar to $p_\text{eq}$. This is manifest from Eqs. (2) and (4): a larger KL between $p_\text{d}$ and $p_\text{eq}$, which is $S_\text{tot}$, suppresses $\mathcal{P}[p_\text{d}]$ further. In practice $p_\text{d}$ is often fixed by the training data and $p_\text{eq}^{(t)}$ changes as we adjust the drift and diffusion coefficients in the forward process, Eq. (12), to speed up the generative process. There is great interest in straightening the trajectories from the Probability Flow (PF) ODE [6] by clever choices of $b_+$ and $\sigma$, to enable few-shot sampling during the generative stage [29–31]. However, such forward processes often produce more entropy, which means these models may inadvertently be placing a higher information load on the neural network.

As an illustrative example, consider the Straight Line Diffusion Model (SL) introduced in [31]. The forward process is

$$\mathrm{d}Y_s = -\frac{1}{1-s}Y_s + \sqrt{\frac{2}{1-s}}\sigma_0 \mathrm{d}B_s. \tag{20}$$

At an intermediate time $s \in (0, T)$ (with $T = 1$), a sample $y_\text{d} \sim p_\text{d}$ is propagated to

$$y_s = (1-s)y_\text{d} + \sigma_0\sqrt{1-(1-s)^2}\epsilon, \tag{21}$$

where $\epsilon \sim \mathcal{N}(0, \mathbb{1}_D)$. For small $\sigma_0$ and a fairly 'wide' $p_\text{d}$, the trajectories in Eq. (21) are nearly straight lines that land at $y_T \sim \mathcal{N}(0, \sigma_0^2 \mathbb{1}_D)$. This is a result of allowing the drift term to dominate the noise in Eq. (20). But that also makes $p_\text{eq}^{(t)} \propto \exp(-y^2/\sigma_0^2)$ a very narrow Gaussian, which increases the KL to $p_\text{d}$ and thereby $S_\text{tot}$. Or, using the intuition from Sec. 2, decreasing the randomness in the diffusion process diminishes the chances of an automatic fluctuation into $p_\text{d}$. Using a forward process with more noise would lower the entropy, but only to a certain extent. If the $\sigma$ is too large $p_\text{eq}^{(t)}$ becomes too wide compared to $p_\text{d}$ and $S_\text{tot}$ rises again.

The above discussion is meant to highlight that $S_\text{tot}$ depends on the forward diffusion process and $p_\text{d}$ in a non-trivial way, and that there might be an optimal process that produces the least entropy for a given $p_\text{d}$. This intuition is made more precise by the *thermodynamic uncertainty relation*, which relates the total entropy produced to the $L^2$-Wasserstein distance between $p_\text{d}$ and $p_0$ [32, 33],

$$S_\text{tot} \times \sigma^2 T \geq \frac{1}{2}\mathcal{W}_2(p_\text{d}, p_0)^2. \tag{22}$$

Here $\sigma$ is assumed to be a constant for simplicity and $\sigma^2 T$ is the time it takes for $p_\text{d}$ to reach $p_0$, measured in units of $\sigma^{-2}$. The $\mathcal{W}_2$ depends only on the initial and final distributions. If two processes take the same time to equilibrate (reach $p_0 \approx p_\text{eq}$), the one whose equilibrium state is farther from $p_\text{d}$ will generate more entropy. If two process transform $p_\text{d}$ to the same $p_0$, the $\mathcal{W}_2$ is the same in both cases, but the faster transformation will produce more entropy to satisfy the bound. Therefore a diffusion model must store more information to reverse a faster diffusion process. This is the thermodynamic speed limit: given $p_\text{d}$ and $p_0$, there is an upper limit to how fast we can diffuse one to the other without exceeding a specific entropy production budget. Equivalently, a faster transformation requires a greater amount of information to reverse, which has been found to affect accuracy [34]. These observations are also confirmed in our experiments.

# 6 Experiments

Neural entropy, as defined in Eq. (11), quantifies the information presented to the network in an idealized setting. In practice, the finiteness of the data, imperfections in training, and strong inductive biases of the network all affect the amount of information stored in the neural network. To address these points we will perform two broad classes of experiments, first to probe the transport properties of diffusion discussed in Eq. (22), and second, to study the storage efficiency of diffusion models.

**Transport experiments**  We work with synthetic datasets sampled from simple multivariate distributions for which we have closed-form expressions for both $p_\text{d}$ and $\nabla \log p$ (e.g. Gaussian mixtures). This allows us to produce as many samples as we require with high fidelity, compute their exact log densities, and work in arbitrary dimensions. Recall from Eq. (17) that the loss function upper bounds the KL divergence between the data distribution and the generated distribution, $\mathrm{D}_{KL}(p_\text{d}(\cdot)\|p_{\boldsymbol{\theta}}(\cdot, T))$,

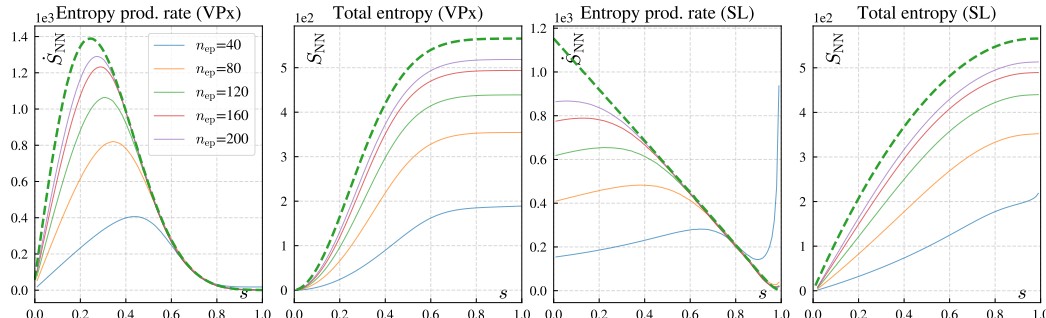

Figure 2: Entropy production rate and total entropy as $p_\mathrm{d}$ is diffused to $p_0$ by the VPx and SL processes from Eq. (24) and Eq. (20) respectively. The dashed lines are the ideal curves for $\dot{S}_\mathrm{tot}$ and $S_\mathrm{tot}$, while the solid lines are $\dot{S}_\mathrm{NN}$ and $S_\mathrm{NN}$ at the end of the $n_\mathrm{ep}$-th training epoch.

which means this KL can be used to assess the performance of the diffusion model—a smaller KL implies the model reproduces $p_\mathrm{d}$ more faithfully. For any sample $x$ we know $p_\mathrm{d}(x)$, so all we need to compute the KL is $\log p_{\boldsymbol{\theta}}(x, T)$. The latter is approximated by the method discussed in App. E.2. Finally, the transport experiments will be carried out with diffusion models with a multi-layer perceptron (MLP) core. Since the inductive biases in such fully connected networks are weak [35], these models enable us to isolate the effects of varying levels of neural entropy on the model performance.

If $p_\mathrm{d}$ is a mixture of Gaussians it is possible to compute Eq. (11) explicitly, which gives the ideal value of neural entropy. However, due to the imperfections mentioned above the actual neural entropy is given by Eq. (18). Both these expressions are computed by Monte Carlo averages, and it is useful to examine how they change over time $s \in (0, T]$. For example, with some new samples $\tilde{y}_\mathrm{d} \sim p_\mathrm{d}$,

$$
S_\mathrm{NN}(s) = \int_0^s \mathrm{d}\bar{s}\, \frac{\sigma^2}{2} \mathbb{E}_p\left[\|\boldsymbol{e}_{\boldsymbol{\theta}}\|^2\right] \approx s\, \mathbb{E}_{\tilde{y}_\mathrm{d} \sim p_\mathrm{d}} \mathbb{E}_{\bar{s} \sim \mathcal{U}(0, s)}\left[\frac{\sigma(\bar{s})^2}{2} \mathbb{E}_{y_{\bar{s}} \sim p(y_{\bar{s}}|\tilde{y}_\mathrm{d})}\left[\|\boldsymbol{e}_{\boldsymbol{\theta}}(y_{\bar{s}}, \bar{s})\|^2\right]\right],
\tag{23}
$$

To explore the considerations raised in Sec. 5 more thoroughly, we experiment with a few different diffusion processes. We introduce a minor generalization of the Variance Preserving (VP) process, given by the SDE

$$
\mathrm{d}Y_s = -\frac{\beta(s)}{2} Y_s\, \mathrm{d}s + \kappa\sqrt{\beta(s)}\mathrm{d}B_s,
\tag{24}
$$

which we shall henceforth refer to as VPx. For $\kappa = 1$ this is the same as the VP process from [1, 6, 29]. If we set $\kappa = \sigma_0$ we obtain a process that has the same quasi-invariant distribution as Eq. (20). However, the trajectories generated by Eq. (24) are

$$
y_s = e^{-\frac{1}{2}\int_0^s \beta(\bar{s})\mathrm{d}\bar{s}} y_\mathrm{d} + \kappa\sqrt{1 - e^{-\int_0^s \beta(\bar{s})\mathrm{d}\bar{s}}}\, \epsilon,
\tag{25}
$$

which 'forgets' $y_\mathrm{d}$ at an exponential rate as opposed to the linear evolution in Eq. (21). This difference is borne out in their respective entropy profiles—plots of the entropy production rate $\dot{S}_\mathrm{tot} \equiv \mathrm{d}S_\mathrm{tot}/\mathrm{d}s$, and the total entropy $S_\mathrm{tot}$, over time. These are the dashed curves in Fig. 2. The final value of $S_\mathrm{tot}$ is very nearly the same for both processes when $\kappa = \sigma_0$ since they have a common $p_\mathrm{eq}^{(t)}$ (cf. Eq. (16)), with small discrepancies arising from differences in their respective $p_0$. The solid lines in Fig. 2 are $\dot{S}_\mathrm{NN}$ and $S_\mathrm{NN}$, evaluated at various stages of training. As we train over more epochs, $n_\mathrm{ep}$, the network absorbs more information, bringing $S_\mathrm{NN}$ closer to its ideal value $S_\mathrm{tot}$.

The experiments in Fig. 2 were carried out on a model trained on $N = 8192$ samples from a mixture of five Gaussians in $D = 6$ dimensions, which is $p_\mathrm{d}$. The Gaussian components were $\mathcal{N}(\bar{x}_r, \mathbb{1}_D)$, where the means $\bar{x}_r$ were randomly chosen from a $D$-hypercube of side length 4 centered at the origin. We used $\kappa = \sigma_0 = 0.1$, so both processes transform $p_\mathrm{d}$ to $p_0 \approx \mathcal{N}(0, 10^{-2}\mathbb{1}_D)$ at $T = 1$ (cf. Eq. (40)). The SL process has a singularity at $T = 1$ exactly (cf. Eq. (20)); this divergence manifests in the $\dot{S}_\mathrm{NN}$ curve for SL at the early stages of training.

In both VPx and SL dynamics, the combination of weak noise with a relatively strong drift subdues the randomness in the diffusion process, leading to larger $S_\mathrm{tot}$ (cf. Sec. 5). Equivalently, the $p_\mathrm{d}$ used above

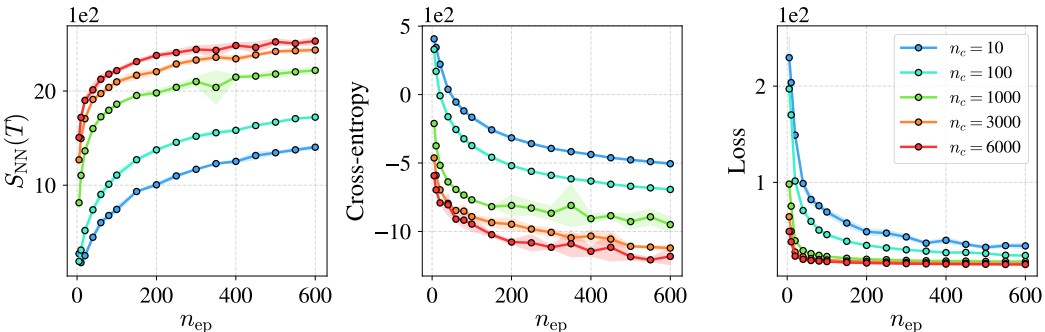

Figure 3: The evolution of neural entropy, cross-entropy, and loss over training epochs for an unconditional image diffusion model (VP) trained on the MNIST dataset. The different colors correspond to models trained on $n_c$ number of samples per class; $n_c = 6000$ means the model was trained on the entire dataset. The growth in neural entropy with the number of samples is nearly logarithmic. The values of $S_{\mathrm{NN}}(T)$ at the end of training are shown in Fig. 16.

is too far from their equilibrium state so a larger amount of information must be supplied to transform $p_{\mathrm{eq}} \approx p_0$ to $p_{\mathrm{d}}$. On the other hand, if we use a regular VP process, for which $p_{\mathrm{eq}} = \mathcal{N}(0, \mathbb{1}_{\mathrm{D}})$, the 'distance' to $p_{\mathrm{d}}$ is smaller and so is the total entropy. As a result, the VP model trains much faster and produces a more accurate reconstruction of $p_{\mathrm{d}}$ than the VPx or SL model. This is shown in Fig. 10.

**Storage experiments** We carry out similar experiments on a simple image diffusion model with a U-net core, trained on the MNIST dataset without class conditioning [36]. In this instance, the training dataset is small relative to the dimensionality of pixel space. Therefore the model relies on the inductive biases of the network to generalize from the given data points rather than memorize them [12]. Entropy curves for this dataset also show a sharp peak in entropy production near $s = 0$ (see Fig. 12). This is because the images live on a manifold $\mathcal{M}_{\mathrm{d}}$ of much lower dimensionality compared to the ambient 784-dimensional space. Seen from the $t$-direction, the sharp rise in entropy as $t \to T$ tells us that the diffusion model needs to inject a lot of information in the final few time steps to locate the sample precisely on $\mathcal{M}_{\mathrm{d}}$. Similar plots for the SL model are given in App. E.

In Fig. 3 we train the image model on the first $n_c$ samples from each class, $N = 10n_c$ samples in total, and measure $S_{\mathrm{NN}}$ and the cross entropy $\mathbb{E}_{p_{\mathrm{d}}}[-\log p_{\boldsymbol{\theta}}]$ as training progresses. We cannot compute the true $S_{\mathrm{tot}}$ or KL since the exact log densities are not known. Nonetheless, we see a similar result as before: the neural entropy and cross-entropy saturates after the model trains for a while. Importantly, the model absorbs more information if it is presented with a larger number of samples, but the growth in $S_{\mathrm{NN}}(T)$ with $N$ is not linear, it appears to be *logarithmic* (see Figs. 1 and 16).

We obtain the same behavior from a diffusion model trained on the CIFAR-10 dataset [37]. This time we use a U-net with self-attention layers [5, 38] and apply class conditioning. The image quality improves substantially with $N$ but neural entropy scales as nearly $\log N$ (see Fig. 18). Notably, such a trend is absent from the Gaussian mixture experiments performed on an MLP-based diffusion model (see Figs. 4 and 15). At low $N$ these models concentrate the probability mass around the sparse data points and learn a very different distribution from the true $p_{\mathrm{d}}$. This is why $S_{\mathrm{NN}}$ is larger at small $N$; it could also be larger than $S_{\mathrm{tot}}$ produced by diffusion of the actual $p_{\mathrm{d}}$. On the other hand, the combination of structured data and well-matched inductive biases in the image models saves us from overfitting even when data density is low, while also compressing the details from additional samples efficiently.

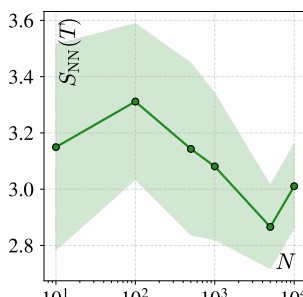

Figure 4: Neural entropy vs. number of samples for a diffusion model with an MLP trained on Gaussian mixtures.

# 7 Conclusion

In Sec. 1 we used the example of storing images to motivate neural entropy. We presented two scenarios: the storage of individual images versus the retention of a distribution of images. In a conventional memory the number of bytes required to store $N$ images will be $N$ times the bytes per image, even with efficient compression. Classical compression algorithms make implicit assumptions about the statistical properties of natural images and apply them universally to all inputs. However a diffusion model sees a large ensemble of images and can *learn* non-local correlations between the pixels of similar groups of images; they are infinitely deep autoencoders [39–41]. In other words, these models can tailor a compression scheme specific to the properties of the data distribution. This is a lossy form of compression of course, since we rarely recover the training images perfectly from these probabilistic models.

How about other generative models? At a conceptual level, Schrödinger's argument can be extended beyond just diffusion. For instance, a very simple-minded approach to creating a sentence of length $L$ would be to choose $L$ words at *random* from a corpus. The vast majority of sentences produced by this process would be utterly meaningless, but once in a while, we get a rare fluctuation that counts as a sensible statement. The collection of all such sentences constitutes an island of meaning $p_{\mathrm{d}}$, in a sea of non-sense $p_{\mathrm{eq}}$. Here $p_{\mathrm{eq}}$ can be a uniform distribution over all $L$-word combinations, say. $\mathcal{P}$ is the distribution of all distributions on these combinations. The information needed to transform $p_{\mathrm{eq}} \rightarrow p_{\mathrm{d}}$ is proportional to the negative logarithm of the probability of fluctuating into $p_{\mathrm{d}}$ automatically. A mechanism that enhances this probability to near certainty would need to store that much information to effect the transformation. If we start with a $p_{\mathrm{eq}}$ that assigns a greater probability to more frequently used words it would be easier, albeit still very improbable, to auto-fluctuate to $p_{\mathrm{d}}$. This is a manifestation of thermodynamic uncertainty, for words.

Diffusion-based LLMs [42, 43] offer the most direct path to defining neural entropy in the context of language. They are based on a discrete diffusion process very similar to the lattice random walks we considered in Secs. 2 and 3. The ideal neural entropy in that case should have a form similar to Eq. (37), with transitions that extend beyond nearest neighbor jumps. The discrete analog of the thermodynamic uncertainty relation, Eq. (22), is discussed in [33]. It would be interesting to investigate how the choice of the forward process in diffusion LLMs affects the training efficiency and model performance. More work will be needed to extend the entropic picture to transformer models, but investigations into the compressive abilities of such models are underway [44].

**Limitations**    Our definition of neural entropy is limited to continuous diffusion models at present. In App. D we also point out the difficulties in defining neural entropy with score-matching models. With respect to the experimental results, the logarithmic growth of neural entropy with $N$ is stated as an empirical observation with little explanation. A deeper investigation of this phenomenon is relegated to future work. In particular, it would be interesting to check whether this trend is repeated in more sophisticated network architectures like diffusion transformers [45], or if there is a connection to the scaling laws [46]. As noted in Figs. 12 and 17 the calculation of neural entropy in the image models requires some care due to divergent entropy production near $s = 0$. This behooves us to investigate the neural entropy in diffusion processes with momentum components which soften such singular behavior [47, 48].

**Related work**    The original work that introduced diffusion models took inspiration from the Jarzynski equality and fluctuation theorem from non-equilibrium thermodynamics [1, 49, 50]. The relation between these ideas becomes apparent through [40, 7], both of which use the Feynman-Kac formula and Girsanov's theorem to develop similar results separately for diffusion models and thermodynamics. These results can also be understood in the language of stochastic optimal control [15, 51]. We illustrate both approaches in App. B.4. Several authors have also developed the connection between these models and the Schrödinger's bridge problem [52, 53]. The consequences of the thermodynamic speed limit on diffusion model accuracy are studied in greater detail in [34]. Information-theoretic aspects of diffusion models are also explored in [54], and a deeper discussion of their ability to capture mutual information is given in [55]. It has also been noted that diffusion models are a form of energy-based memory [56], which concurs with the discussion in Sec. 4 and App. C.2, since the generative process in these models is a descent along a learned free energy landscape.

## Acknowledgments and Disclosure of Funding

We thank Lorenzo Orecchia, William Cottrell, Austin Joyce, Maurice Weiler, and Ryan Robinett for valuable discussions, and Yuji Hirono for bringing [34] to our attention. We are especially grateful to William Cottrell and Lorenzo Orecchia for their comments on an earlier version of this paper. The author was supported in part by the Kavli Institute for Cosmological Physics at the University of Chicago through an endowment from the Kavli Foundation. Computational resources for this project were made available through the AI+Science research funding from the Data Science Institute at the University of Chicago. Code is available at https://github.com/akhilprem1/NeuralEntropy

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

# Contents

# A   Random walk on a lattice

Random walks on a discrete one-dimensional lattice serve as a simple toy model to understand many of the results in this paper. Consider $M$ random walkers on a lattice of spacing $\ell$, each of whom can occupy any of the sites $x = n\ell, n \in \mathbb{Z}$. We shall use the time variable $s$ for the forward evolution of the walkers. In every time step $\Delta s$ every walker at $x$ jumps either left or right with probability $q_L(x, s)$ and $q_R(x, s)$ respectively, so $q_L + q_R = 1$. Then, the probability of finding a walker at $x$ updates as

$$p(x, s + \Delta s) = q_R(x - \ell, s)p(x - \ell, s) + q_L(x + \ell, s)p(x + \ell, s), \tag{26}$$

since all walkers that were at $x$ at time $s$ cleared out and have been replaced by incoming walkers from either the left or right. Taylor expanding Eq. (26) in small $\Delta s$ and $\ell$,

$$\partial_s p(x, s) = -\frac{\ell}{\Delta s}\partial_x((q_R(x, s) - q_L(x, s))p(x, s)) + \frac{\ell^2}{2\Delta s}\partial_x^2 p(x, s), \tag{27}$$

which is the Fokker-Planck equation with drift and diffusion coefficients

$$b_+(x, s) = \frac{\ell}{\Delta s}(q_R(x, s) - q_L(x, s)), \quad \sigma^2 = \frac{\ell^2}{\Delta s}. \tag{28}$$

Conversely, given a Fokker-Planck equation with a generic drift $b_+$ we can think of it as the small $\ell$ limit of a lattice model with jump probabilities

$$q_R(x, s) = \frac{1}{2} + \frac{\Delta s}{2\ell}b_+(x, s), \quad q_L(x, s) = \frac{1}{2} - \frac{\Delta s}{2\ell}b_+(x, s). \tag{29}$$

If the diffusion coefficient is time and/or position dependent we can map it to Eq. (28) by rescaling time and/or by a change of variables [57].

## A.1   Reversal

Eq. (26) can be reversed by returning to $x - \ell$ and $x + \ell$ respectively a fraction

$$\frac{q_R(x - \ell, s)p(x - \ell, s)}{p(x, s + \Delta s)} =: \tilde{q}_L(x, s + \Delta s), \tag{30a}$$

$$\frac{q_L(x + \ell, s)p(x + \ell, s)}{p(x, s + \Delta s)} =: \tilde{q}_R(x, s + \Delta s) \tag{30b}$$

of $p(x, s + \Delta s)$. It is useful to introduce a new time variable $t$ for the reverse direction, as shown in Fig. 5. Then, the site $x$ receives $\tilde{q}_R$ fraction of the contents of $x - \ell$, and $\tilde{q}_L$ of the walkers in $x + \ell$ from time $t$, and the distribution of walkers evolve as

$$\tilde{p}(x, t + \Delta t) = \tilde{q}_R(x - \ell, t)\tilde{p}(x - \ell, t) + \tilde{q}_L(x + \ell, t)\tilde{p}(x + \ell, t), \tag{31}$$

In the small $\ell$ limit, we can compute the new drift (cf. Eq. (41)),

$$
\begin{aligned}
b_-(x, t) &:= \frac{\ell}{\Delta t}(\tilde{q}_L(x, t) - \tilde{q}_R(x, t)) \\
&= \frac{\ell}{\Delta t}(q_R - q_L) - \frac{\ell^2}{\Delta t}\partial_x \log p + \frac{1}{p}\frac{\ell^3}{2\Delta t}\partial_x^2((q_R - q_L)p) - \ell(q_R - q_L)\partial_s p + \dots \Big|_{x,s} \\
&= b_+(x, s) - \sigma(x, s)^2 \partial_x \log p(x, s) + \mathcal{O}(\ell^3). \tag{32}
\end{aligned}
$$

The diffusion coefficient remains unchanged at the same order,

$$\frac{\ell^2}{\Delta t}(\tilde{q}_R(x, t) + \tilde{q}_L(x, t)) = \frac{\ell^2}{\Delta t}(q_R(x, t) + q_L(x, t)) + \mathcal{O}(\ell^3). \tag{33}$$

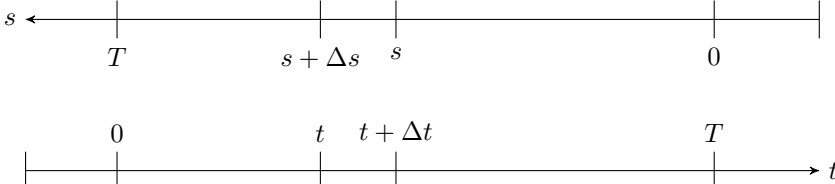

Figure 5: Time variables in the forward (top) and reverse (bottom) directions.

## A.2 Entropy production

Eq. (31) is not the only way to return to the original distribution of walkers, but it is the strategy that corresponds to the optimal kernel $h_\star$ in Eq. (3). In that expression, $g$ is the transition kernel for the forward dynamics Eq. (26), now applied in the $t$-direction. Explicitly,

$$h_\star(x_{t+\Delta t}|x_t) = \tilde{q}_R(x_t,t)\delta_{x_{t+\Delta t},x_t+\ell} + \tilde{q}_L(x_t,t)\delta_{x_{t+\Delta t},x_t-\ell}, \tag{34a}$$

$$g(x_{t+\Delta t}|x_t) = q_R(x_t,t)\delta_{x_{t+\Delta t},x_t+\ell} + q_L(x_t,t)\delta_{x_{t+\Delta t},x_t-\ell}. \tag{34b}$$

Here $\delta_{x_{t+\Delta t},x_t+\ell}$ means $x_{t+\Delta t}$ is to the right of $x_t$ etc. We can compute Eq. (3) explicitly for the random walker on a lattice.

$$D_{KL}(h_\star\|g) = \sum_{x_0,x_T} p_{\text{eq}}(x_0)h_\star(x_T|x_0)\log\frac{h_\star(x_T|x_0)}{g(x_T|x_0)}. \tag{35}$$

Using the log sum inequality,

$$D_{KL}(h_\star\|g) \leq \sum_{t=0}^{T-\Delta t}\sum_{x_{t+\Delta t},x_t} p(x_t,t)h_\star(x_{t+\Delta t}|x_t)\log\frac{h_\star(x_{t+\Delta t}|x_t)}{g(x_{t+\Delta t}|x_t)} =: S_{\text{tot}}. \tag{36}$$

Using Eq. (34),

$$S_{\text{tot}} = \sum_{t=0}^{T-\Delta t}\sum_{x_t} p(x_t,t)\left(\tilde{q}_R\log\frac{\tilde{q}_R}{q_R} + \tilde{q}_L\log\frac{\tilde{q}_L}{q_L}\right)\bigg|_{x_t,t}. \tag{37}$$

Substituting Eq. (30) and Taylor expanding,

$$S_{\text{tot}} = \sum_{t=0}^{n-1}\sum_{x_t} p(q_L - q_R)\log\frac{q_L}{q_R} + \ell\partial_x p\log\frac{q_L}{q_R} + \frac{\ell^2}{2}p\left(q_L(\partial_x\log(q_L p))^2 + q_R(\partial_x\log(q_R p))^2\right)$$

$$+ \ell\partial_x((q_L - q_R)p) + \frac{\ell^2}{2}\partial_x^2 p - \Delta s\left(\frac{q_L}{q_R}\partial_s(q_R p) + \frac{q_R}{q_L}\partial_s(q_L p)\right) + \mathcal{O}(\ell^3). \tag{38}$$

Expressing the jump probabilities in terms of $b_+$ and $\sigma^2$ (cf. Eq. (29)) and using the Fokker-Planck equation to eliminate some terms, we obtain the final expression for total entropy [16]:

$$S_{\text{tot}} = \int_0^T dt\frac{1}{2\sigma^2}\mathbb{E}_p\left[\left\|2b_+ - \sigma^2\partial_x\log p\right\|^2\right]. \tag{39}$$

# B  Stochastic control

## B.1  Notation

We use the time variable $s$ for the forward diffusion process, which runs from right ($s = 0$) to left ($s = T$) in Fig. 6. Sometimes we indicate functions of $s$ as $\overleftarrow{f}$ to remove ambiguity when the same function is also expressed in terms of time variable $t = T - s$. That is, $\overleftarrow{f}(s) = \overleftarrow{f}(T-t) = f(t)$. For instance, $p(x,t) \equiv \overleftarrow{p}(x,s)$. $\hat{B}_s$ and $B_t$ denote the Brownian motions associated with the forward and reverse/controlled SDEs, respectively. $\nabla$ is the gradient with respect the spatial coordinates, and $\partial_t, \partial_s$ are partial time derivatives. $S_{\text{tot}}$ is the total entropy produced during forward diffusion,

$S_{\mathrm{G}}$ is the non-equilibrium Gibbs entropy of the distribution, and $S_{\mathrm{NN}}$ is the neural entropy. We make the time-dependence of the entropies explicit later in the paper after we have introduced the $s$ variable; $S_{\mathrm{tot}}$ and $S_{\mathrm{NN}}$ without the time argument should be understood as $S_{\mathrm{tot}}(s = T) \equiv S_{\mathrm{tot}}(T)$. Throughout the paper, we set Boltzmann's constant to unity, $k_{\mathrm{B}} = 1$. log is the natural logarithm. $p_{\mathrm{d}}$ and $p_0$ denote the initial ($s = 0$) and final ($s = T$) densities for the forward process, and $p_{\mathrm{eq}}$ is its equilibrium state. Diffusion takes an infinite time to equilibrate but we always take $T$ to be large compared to the intrinsic time scale of the diffusion process, which is why we ignore the difference between $p_0$ and $p_{\mathrm{eq}}$ in Secs. 2 and 3. $p_u(\cdot, 0)$ and $p_u(\cdot, T)$ are the initial ($t = 0$) and final ($t = T$) densities of the controlled process. There is a slight abuse of notation here because $p_u(\cdot, 0)$ is a distribution that does not depend on the control $u$, it is just the initial state to which the control is applied.

**Assumptions**    We make the same assumptions given in App. A of [20], with the following additions for entropy-matching models:

1. $\exists C > 0 \, \forall x \in \mathbb{R}^D, t \in [0, T] : \|e_{\boldsymbol{\theta}}(x, t)\|_2 < C(1 + \|x\|_2)$,
2. $\exists C > 0 \, \forall x, y \in \mathbb{R}^D, t \in [0, T] : \|e_{\boldsymbol{\theta}}(x, t) - e_{\boldsymbol{\theta}}(y, t)\|_2 < C\|x - y\|_2$,
3. Novikov's condition: $\mathbb{E}_p \left[ \exp \left( \int_0^T \mathrm{d}t \, \frac{1}{2} \left\| \nabla \log p_{\mathrm{eq}}^{(t)} - \nabla \log p + e_{\boldsymbol{\theta}} \right\|^2 \right) \right] < \infty$.

## B.2   A fluctuation relation for diffusion models

Given a set of data vectors, probabilistic models attempt to learn the underlying data distribution from which these vectors could have been sampled. One way to do this is to minimize the KL divergence between the data and the model distributions. Score-based diffusion models are trained by optimizing an objective that upper bounds this KL [20, 40]. In this section, we extend this bound to a more general parameterization of the generative process.

Consider a $D$-dimensional probability density function $p_{\mathrm{d}}$ subjected to a diffusive process

$$\mathrm{d}Y_s = b_+(Y_s, s)\mathrm{d}s + \sigma(s)\mathrm{d}\hat{B}_s. \tag{40}$$

The noise is isotropic and position-independent. Under Eq. (40), the distribution $p_{\mathrm{d}}(y) \equiv \overleftarrow{p}(y, 0)$ evolves to some another distribution $p_0(y) \equiv \overleftarrow{p}(y, T)$ (see Fig. 6). This process can be reversed by an SDE [58–61]

$$\mathrm{d}X_t = -b_-(X_t, T - t)\mathrm{d}t + \sigma(T - t)\mathrm{d}B_t, \tag{41}$$

where $t = T - s$, and the drift term is

$$b_-(X, s) := b_+(X, s) - \sigma^2(s)\nabla \log \overleftarrow{p}(X, s). \tag{42}$$

Starting from $p_0(x) \equiv p(x, 0)$, the reverse evolution back to $p_{\mathrm{d}}(x) \equiv p(x, T)$ appears as a playback of the forward process in Eq. (40), so that $p(x, t) = \overleftarrow{p}(x, T - t)$ at an intermediate time $t$. Crucially, we need information about the forward process, specifically the *score function* $\nabla \log \overleftarrow{p}$, to construct the reverse drift term in Eq. (42). This makes sense: the final distribution $p_0$ has little to no memory of the initial state $p_{\mathrm{d}}$, meaning that many different $p_{\mathrm{d}}$ could diffuse to roughly the same $p_0$, rendering the problem non-invertible without explicit knowledge of the forward process.

If we replace $b_-$ in Eq. (41) with a different drift term $u$, called the *control*, and evolve $p_0(x)$ by the stochastic process

$$\mathrm{d}X_t = -u(X_t, t)\mathrm{d}t + \sigma(T - t)\mathrm{d}B_t, \tag{43}$$

the density $p_u(x, t)$ of $X_t$ will differ from $p(x, t)$, and land on a terminal distribution $p_u(x, T) \neq p(x, T)$. The KL divergence between these distributions is bounded as

$$\int_0^T \mathrm{d}t \, \frac{1}{2\sigma^2} \mathbb{E}_{p(\cdot, t)} \left[ \|b_- - u\|^2 \right] \geq \mathrm{D}_{KL}\left( p(\cdot, T) \| p_u(\cdot, T) \right). \tag{44}$$

More generally, if we start at some $p_u(x, 0) \neq p_0(x)$,

$$\int_0^T \mathrm{d}t \, \frac{1}{2\sigma^2} \mathbb{E}_p \left[ \|b_- - u\|^2 \right] + \mathrm{D}_{KL}\left( p_0(\cdot) \| p_u(\cdot, 0) \right) \geq \mathrm{D}_{KL}\left( p(\cdot, T) \| p_u(\cdot, T) \right). \tag{45}$$

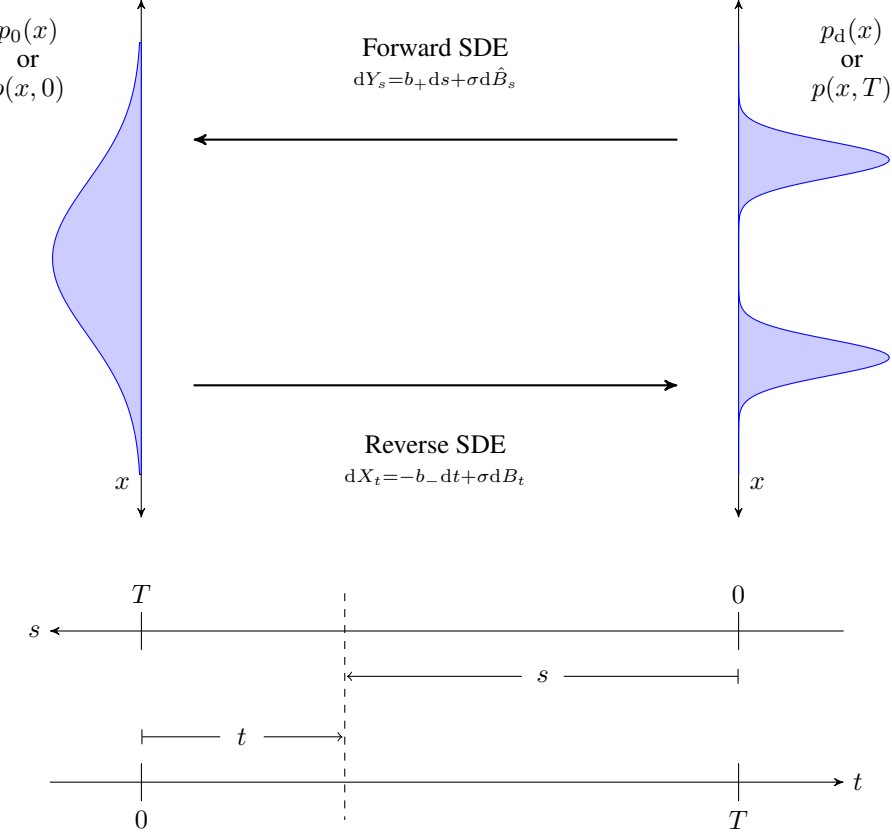

Figure 6: A schematic of the forward and reverse diffusion processes.

This result can be derived using the theory of stochastic optimal control [15] or by an application of the Feynman-Kac formula and Girsanov's theorem [40, 7]. The details are given in App. B.4. See also [62].

As a particular example, we can choose $u = b_+ - \sigma^2 s_{\boldsymbol{\theta}}$, where $s_{\boldsymbol{\theta}}$ is the output of a neural network, which converts the l.h.s. in Eq. (44) into the score-matching objective from [5, 6]. This leads to Theorem 1 from [20] (cf. Eq. (19)),

$$\int_0^T dt\, \frac{\sigma^2}{2} \mathbb{E}_p \left[ \|s_{\boldsymbol{\theta}} - \nabla \log p\|^2 \right] + \mathrm{D}_{KL}\left(p_0(\cdot)\|p_{\boldsymbol{\theta}}(\cdot, 0)\right) \geq \mathrm{D}_{KL}\left(p(\cdot, T)\|p_{\boldsymbol{\theta}}(\cdot, T)\right). \quad (46)$$

If we pick the initial distribution $p_{\boldsymbol{\theta}}(x, 0)$ to be close to $p_0(x)$, and train a neural network to minimize the score-matching term, we can tighten the KL divergence between the data distribution, $p(x, T) \equiv p_{\mathrm{d}}(x)$, and the generated one, $p_{\boldsymbol{\theta}}(x, T)$. This is how score-matching diffusion models work. Similarly, the parameterization $u = -b_+ - \sigma^2 e_{\boldsymbol{\theta}}$ gives the entropy-matching objective (cf. Eq. (17)),

$$\int_0^T dt\, \frac{\sigma^2}{2} \mathbb{E}_p \left[ \left\| \frac{2b_+}{\sigma^2} - \nabla \log p + e_{\boldsymbol{\theta}} \right\|^2 \right] + \mathrm{D}_{KL}\left(p_0(\cdot)\|p_{\boldsymbol{\theta}}(\cdot, 0)\right) \geq \mathrm{D}_{KL}\left(p(\cdot, T)\|p_{\boldsymbol{\theta}}(\cdot, T)\right). \quad (47)$$

Lastly, if we set $u = -b_+$ and choose $p_u(x, 0) = p_0(x)$ for simplicity, we obtain a lower bound on the total entropy (cf. Eq. (16))

$$S_{\mathrm{tot}}(T) \equiv \int_0^T dt\, \frac{\sigma^2}{2} \mathbb{E}_p \left[ \left\| \frac{2b_+}{\sigma^2} - \nabla \log p \right\|^2 \right] \geq \mathrm{D}_{KL}\left(p(\cdot, T)\|p_{b_+}(\cdot, T)\right). \quad (48)$$

Next, we look at the conditions for which this bound is saturated.

## B.3 The $H$-theorem

We derive Eq. (15) here, following [15]. It states that $p$ relaxes toward $p^{\text{eq}}$, with the rate of approach slowing down as it nears that state. To prove it we start with the time derivative

$$-\frac{\mathrm{d}}{\mathrm{d}t}\mathrm{D}_{KL}\left(p(x,t)\|p_t^{\text{eq}}(x)\right)$$

$$=\frac{\mathrm{d}}{\mathrm{d}t}\mathbb{E}_p\left[-\log\frac{p(x,t)}{p_t^{\text{eq}}(x)}\right]$$

$$=\frac{\mathrm{d}}{\mathrm{d}t}\left(-\int\mathrm{d}x\,p(x,t)\log p(x,t)\right)-\frac{\mathrm{d}}{\mathrm{d}t}\left(-\int\mathrm{d}x\,p(x,t)\log p_t^{\text{eq}}(x)\right). \quad (49)$$

The first term on the r.h.s. is the Gibbs entropy production rate $\dot{S}(t)$ [9],

$$\dot{S}(t)=-\int\mathrm{d}x\,\partial_t p\log p-\int\mathrm{d}x\,\partial_t p$$

$$=\int\mathrm{d}x\left(-\nabla\cdot(b_-p)-\frac{\sigma^2}{2}\nabla^2 p\right)\log p+\int\mathrm{d}x\nabla(\cdots)$$

$$\overset{\text{IBP}}{=}\int\mathrm{d}x\,p(x,t)\left(\frac{\sigma^2}{2}\|\nabla\log p\|^2+b_-\cdot\nabla\log p\right). \quad (50)$$

The second term in Eq. (49) can be simplified by using the Fokker-Planck equations for $p$ and $p^{\text{eq}}$,

$$\frac{\mathrm{d}}{\mathrm{d}t}\left(\int\mathrm{d}x\,p(x,t)\log p_t^{\text{eq}}(x)\right)$$

$$=\int\mathrm{d}x\left(\partial_t p(x,t)\log p_t^{\text{eq}}(x)+p(x,t)\frac{\partial_t p_t^{\text{eq}}(x)}{p_t^{\text{eq}}(x)}\right)$$

$$=\int\mathrm{d}x\left(\nabla\cdot(b_-p)+\frac{\sigma^2}{2}\nabla^2 p\right)\log p_t^{\text{eq}}(x)+\int\mathrm{d}x\frac{p(x,t)}{p_t^{\text{eq}}(x)}\left(-\nabla\cdot(b_+p_t^{\text{eq}})+\frac{\sigma^2}{2}\nabla^2 p_t^{\text{eq}}\right).$$

Here, we used the fact that $\partial_t p_t^{\text{eq}}=0$ when $b_+/\sigma^2=\text{const.}$, and replaced that zero with Eq. (14). We will consider each new term separately for clarity. Integrating by parts,

$$\int\mathrm{d}x\left(\nabla\cdot(b_-p)+\frac{\sigma^2}{2}\nabla^2 p\right)\log p_t^{\text{eq}}(x)$$

$$=\int\mathrm{d}x\,p(x,t)\left(-b_-\cdot\nabla\log p_t^{\text{eq}}-\frac{\sigma^2}{2}\nabla\log p\cdot\nabla\log p_t^{\text{eq}}\right),$$

and

$$\int\mathrm{d}x\frac{p(x,t)}{p_t^{\text{eq}}(x)}\left(-\nabla\cdot(b_+p_t^{\text{eq}})+\frac{\sigma^2}{2}\nabla^2 p_t^{\text{eq}}\right)$$

$$=\int\mathrm{d}x\left(b_+\cdot\nabla p-b_+p\cdot\frac{\nabla p_t^{\text{eq}}}{p_t^{\text{eq}}}-\frac{\sigma^2}{2}\nabla\log p\cdot\nabla\log p_t^{\text{eq}}+\frac{\sigma^2}{2}p\|\nabla\log p_t^{\text{eq}}\|^2\right),$$

$$=\int\mathrm{d}x\,p(x,t)\left(b_+\cdot(\nabla\log p-\nabla\log p_t^{\text{eq}})-\frac{\sigma^2}{2}\nabla\log p\cdot\nabla\log p_t^{\text{eq}}+\frac{\sigma^2}{2}\|\nabla\log p_t^{\text{eq}}\|^2\right).$$

Adding up everything, we obtain

$$-\frac{\mathrm{d}}{\mathrm{d}t}\mathrm{D}_{KL}\left(p(x,t)\|p_t^{\text{eq}}(x)\right)$$

$$=\mathbb{E}_p\left[\frac{\sigma^2}{2}\|\nabla\log p(x,t)-\nabla\log p_t^{\text{eq}}(x)\|^2+(b_-+b_+)\cdot(\nabla\log p(x,t)-\nabla\log p_t^{\text{eq}}(x))\right]$$

$$=-\mathbb{E}_p\left[\frac{1}{2\sigma^2}\|2b_+-\sigma^2\nabla\log p(x,t)\|^2\right], \quad (51)$$

where in the last step we used Eq. (14) to replace $\nabla\log p_t^{\text{eq}}(x)$. Integrating Eq. (51) yields Eq. (15).

## B.4 Derivation of the bound

We present a derivation of the bound in Eq. (44), drawing from the proofs in [40, 15]. We only outline the steps here, and refer the reader to those papers for more formal details. Consider the process specified by the SDE

$$\mathrm{d}X_t = v(X_t, t)\mathrm{d}t + \sigma(T-t)\mathrm{d}B_t \tag{52}$$

with the initial condition $X_0 \sim p_v(\cdot, 0)$. The evolution of $p_v(x, t)$ under Eq. (52) is given by the Fokker-Planck equation

$$\partial_t p_v + \nabla \cdot (v p_v) - \frac{\sigma^2}{2}\nabla^2 p_v = 0. \tag{53}$$

Switching the time variable to $s = T - t$ (see Fig. 6) converts this into a backward Kolmogorov equation for $\overleftarrow{p}_v(\cdot, s) := p_v(\cdot, t)$,

$$\partial_s \overleftarrow{p}_v - (\nabla \cdot v)\overleftarrow{p}_v - v \cdot \nabla \overleftarrow{p}_v + \frac{\sigma^2}{2}\nabla^2 \overleftarrow{p}_v = 0, \tag{54}$$

with the terminal condition $\overleftarrow{p}_v(\cdot, T) = p_v(\cdot, 0)$. The solution for Eq. (54) is given by the *Feynman-Kac formula* [63],

$$\overleftarrow{p}_v(x, s) = \mathbb{E}\left[\overleftarrow{p}_v(Y_T, T)\exp(-\int_s^T \mathrm{d}\bar{s}\,\nabla \cdot v(Y_{\bar{s}}, T-\bar{s}))\Big|Y_s = x\right], \tag{55}$$

where $Y_{\bar{s}}$ is a diffusion process that solves

$$\mathrm{d}Y_s = -v(Y_s, T-s)\mathrm{d}s + \sigma(s)\mathrm{d}B_s'. \tag{56}$$

That is, Eq. (55) is a *path integral* over all paths that start from $x$ at time $s$ and evolve under Eq. (56). Setting $s = 0$ in Eq. (55) gives us the likelihood $p_v(\cdot, T) \equiv \overleftarrow{p}_v(\cdot, 0)$. Next, [40] bounds the log likelihood by a change of measure and Jensen's inequality,

$$\log p_v(x, T) \geq \mathbb{E}_{\mathbb{Q}}\left[\frac{\mathrm{d}\mathbb{P}}{\mathrm{d}\mathbb{Q}} + \log p_v(Y_T, 0) - \int_0^T \mathrm{d}\bar{s}\,\nabla \cdot v\Big|Y_0 = x\right]. \tag{57}$$

Here $\frac{\mathrm{d}\mathbb{P}}{\mathrm{d}\mathbb{Q}}$ is the Radon-Nikodym derivative. For our purposes it is enough to understand the expectation value of this object as

$$\mathbb{E}_{\mathbb{Q}}\left[\frac{\mathrm{d}\mathbb{P}}{\mathrm{d}\mathbb{Q}}\Big|Y_0 = x\right] = -\int \mathrm{d}y_T\,Q(y_T|x)\log\frac{Q(y_T|x)}{P(y_T|x)}, \tag{58}$$

where $P$ is the transition probability corresponding to Eq. (56) and $Q$ is the transition probability for a new process[1]

$$\mathrm{d}Y_s = w(Y_s, s)\mathrm{d}s + \sigma(s)\mathrm{d}\hat{B}_s. \tag{59}$$

Then, Eq. (58) can be simplified [14], allowing the r.h.s. in Eq. (57) to be written as the negative of a cost functional (at $s = 0$)

$$\overleftarrow{J}(x, s; v, w) := \mathbb{E}_w\left[\int_s^T \mathrm{d}\bar{s}\left(\frac{1}{2\sigma^2}\|v + w\|^2 + \nabla \cdot v\right) - \log p_v(Y_T, 0)\Big|Y_s = x\right]. \tag{60}$$

We have tinkered with the notation a little, using $\mathbb{E}_w$ to indicate that the averages are taken over Eq. (59). Eqs. (57) and (60) will be used in two different ways below. We will set $p_v(\cdot, 0) = p_0(\cdot)$ in both cases for simplicity.

**Case 1:** $v = -u$, $w = b_+$. Then, Eq. (52) becomes the controlled process Eq. (43) and Eq. (59) is the forward diffusion from Eq. (40), and Eq. (60) is

$$\log p_u(x, T) \geq -\overleftarrow{J}(x, 0; -u, b_+). \tag{61}$$

---

[1]$B_s'$ is a reparameterization of $\hat{B}_s$. See Sec. 4 of [40]

**Case 2:** $v = -b_-$. Under this choice Eq. (52) is the reverse diffusion process from Eq. (41), which takes $p_0 \to p_\mathrm{d}$ via $p$. Then,

$$\log p(x, T) \geq -\overleftarrow{J}(x, 0; -b_-, w). \tag{62}$$

This inequality is saturated if we set $w = b_+$ [15]. To see this, we define the *value function* $\overleftarrow{W}(x, s) := \min_w \overleftarrow{J}(x, s; -b_-, w)$, which is the minimum cost over all admissible values of $w$. It satisfies the *Dynamic Programming equation* [64, 65],

$$\partial_s \overleftarrow{W} + \frac{\sigma^2}{2} \nabla^2 \overleftarrow{W} - \nabla \cdot b_- = \min_w \left( -\frac{1}{2\sigma^2} \|b_- - w\|^2 - w \cdot \nabla \overleftarrow{W} \right). \tag{63}$$

Pointwise minimization of the r.h.s. gives $w_\star = b_- - \sigma^2 \nabla \overleftarrow{W}$. Substituting this back into Eq. (63) we find that $\overleftarrow{W}$ solves

$$\partial_s \overleftarrow{W} + b_+ \cdot \nabla \overleftarrow{W} + \frac{\sigma^2}{2} \nabla^2 \overleftarrow{W} = \frac{\sigma^2}{2} \|\nabla \overleftarrow{W}\|^2 + \sigma^2 \nabla \log \overleftarrow{p} \cdot \nabla \overleftarrow{W} + \nabla \cdot b_-, \tag{64}$$

with terminal value $\overleftarrow{W}(x, T) = -\log p_0(x)$. But notice that, for $v = -b_-$, Eq. (54) can be written as following equation for $\overleftarrow{S} := -\log \overleftarrow{p}$,

$$\partial_s \overleftarrow{S} + b_+ \cdot \nabla \overleftarrow{S} + \frac{\sigma^2}{2} \nabla^2 \overleftarrow{S} = -\frac{\sigma^2}{2} \|\nabla \overleftarrow{S}\|^2 + \nabla \cdot b_-, \tag{65}$$

also with a terminal value $\overleftarrow{S}(x, T) = -\log p_0(x)$. Comparing Eqs. (64) and (65), we see that $\overleftarrow{W}(x, s) = -\log \overleftarrow{p}(x, s)$, and Eq. (62) becomes

$$
\begin{aligned}
\log p(x, T) &= -\overleftarrow{W}(x, 0) \\
&= -\mathbb{E}_{b_+} \left[ \int_s^T d\bar{s} \left( \frac{1}{2\sigma^2} \|\nabla \log \overleftarrow{p}\|^2 - \nabla \cdot b_- \right) - \log p_0(Y_T) \middle| Y_0 = x \right].
\end{aligned} \tag{66}
$$

The bound on the KL divergence between $p$ and $p_u$ can be obtained by integrating Eqs. (61) and (66) over $p_\mathrm{d}$,

$$-\int_0^T d\bar{s}\, \mathbb{E}_{\overleftarrow{p}} \left[ \frac{\|b_+ - b_-\|^2 - \|b_+ - u\|^2}{2\sigma^2} - \nabla \cdot (b_- - u) \right] \geq \mathrm{D}_{KL}\left( p(\cdot, T) \| p_u(\cdot, T) \right). \tag{67}$$

The last term in the average can be integrated by parts,

$$-\mathbb{E}_{\overleftarrow{p}}\left[ \nabla \cdot (b_- - u) \right] = \mathbb{E}_{\overleftarrow{p}}\left[ (b_- - u) \cdot \nabla \log \overleftarrow{p} \right] \overset{(42)}{=} \frac{1}{\sigma^2} \mathbb{E}_{\overleftarrow{p}}\left[ (b_- - u) \cdot (b_- - b_+) \right], \tag{68}$$

to rewrite Eq. (67) in its final form

$$\int_0^T d\bar{s}\, \frac{1}{2\sigma^2} \mathbb{E}_{\overleftarrow{p}}\left[ \|b_- - u\|^2 \right] \geq \mathrm{D}_{KL}\left( p(\cdot, T) \| p_u(\cdot, T) \right). \tag{69}$$

Note that (a) since $p(\cdot, t) = \overleftarrow{p}(\cdot, s)$ we can replace the average $\mathbb{E}_{\overleftarrow{p}} \to \mathbb{E}_p$ and change the time integral to run over $t$, which gives Eq. (44), and (b) we would have an additional KL term if we had $p_0(\cdot) \neq p_u(\cdot, 0)$ in Eq. (67), the one from Eq. (45).

## C   Non-equilibrium thermodynamics

### C.1   Dissipation, lag, and the information gap

Eq. (15) is a Jarzynski equality, applied to Langevin dynamics. Assuming $p_0 \approx p_\mathrm{eq}^{(0)}$ so that we can ignore the KL between them, Eq. (15) is

$$S_\mathrm{tot} = \mathrm{D}_{KL}\left( p_\mathrm{d} \middle\| p_\mathrm{eq}^{(T)} \right). \tag{70}$$

The l.h.s. is the total entropy produced as the distribution $p_d$ is diffused to $p_0$ by Eq. (40) [9]. As diffusion progresses in the $s$-direction, our knowledge of the system diminishes over time. $S_{\text{tot}}$ is a measure of this information loss. Seen from the $t$-direction, Eq. (41) restores the information worn away in the forward process. This is the perspective put forward in the Vaikuntanathan-Jarzynski (VJ) relation for irreversible processes [27],

$$W_{\text{diss}}(t) = \beta^{-1} \text{D}_{KL}(\rho_t \| \rho_t^{\text{eq}}). \tag{71}$$

This equation can be understood by considering a system, initially at a temperature $\beta^{-1}$, driven away from equilibrium by varying an external parameter $\lambda$ from $A$ to $B$, over a time interval $t \in [0, T]$. Let $\langle W \rangle$ be the average mechanical work needed to effect this transformation which, according to the Second Law, is at least equal to the free energy difference $\Delta F$ between $A$ and $B$. Then, $W_{\text{diss}} = \langle W \rangle - \Delta F$ is the average work dissipated over the whole process. $\rho_t$ is the phase space density as the system evolves from $A$ to $B$, and $\rho_t^{\text{eq}}$ is the equilibrium density corresponding to the value of the parameter at that instant, $\lambda_t$.[2] That is, if we adjust the parameter to $\lambda_t$ and wait a long time, the system will evolve to $\rho_t^{\text{eq}}$, its entropy increasing monotonically during the process. This is Boltzmann's $H$-theorem [66, 15]. In other words, $\rho_t^{\text{eq}}$ is the maximum entropy (or minimum information) distribution consistent with $\lambda_t$ [3, 4].

On the other hand, under finite time non-equilibrium evolution the system is rushed along to the state $\rho_t$ and is not afforded the time to relax to the maximum entropy configuration. As a result, a *lag* develops between $\rho_t$ and $\rho_t^{\text{eq}}$, as measured by the KL divergence in Eq. (71). Lag indicates the extent to which the system is out of equilibrium. The VJ relation, Eq. (71), says that the dissipated work dictates the maximum extent to which the equilibrium can be broken at a given instant.

We can also interpret the lag as the *information gap* between $\rho_t$ and $\rho_t^{\text{eq}}$. The entropy of a system is a measure of missing information, with larger entropy associated with a greater degree of ignorance about the system's true state. $\rho_t^{\text{eq}}$ has a higher entropy than $\rho_t$ since much of the information in the latter is lost when $\rho_t$ equilibrates to $\rho_t^{\text{eq}}$. Intuitively, it is clear that the gap is precisely the amount of information we need to exhume $\rho_t$ from $\rho_t^{\text{eq}}$.

In the context of Eq. (70), the non-equilibrium process is the reverse diffusion from Eq. (41). The VJ relation forces a shift in perspective, nudging us to think of Eq. (13b) as the 'native' dynamics of the system, with the process in Eq. (13a) driving it away from its 'preferred state' $p_{\text{eq}}^{(t)}$. The gap measures the information deficit that keeps the native dynamics from reaching $p_d$ on its own. This is also the lesson from Schrödinger's thought experiment (cf. Sec. 2). We can close the gap by modifying the native dynamics to enhance the probability of the outcome $p_d$ (cf. Eq. (9)). The *maximum* additional information needed to do this is $S_{\text{tot}}$.

These arguments also serve to illustrate a specific point about the Second Law: it is a statement about the irreversibility of a non-equilibrium transformation, *even if* that process is simulated on a computer. In the real world irreversibility is an observed property of almost all physical processes[3] [68, 69], but we take this for granted since we evolve with SDEs that are not time-symmetric by construction (cf. Eq. (40)).

## C.2 Entropy and free energy

Shannon entropy can be understood as a measure of ignorance. The analogous quantity we use for diffusive processes is the non-equilibrium Gibbs entropy [9]

$$S_{\text{G}}[\overleftarrow{p}] := -\int \mathrm{d}x \, \overleftarrow{p}(x, s) \log \overleftarrow{p}(x, s). \tag{72}$$

$S_{\text{G}}$ can be understood as a continuous version of the Shannon entropy,[4] up to a multiplicative factor $k_{\text{B}}$ which we set to 1 (see footnote 15 in [3]). Gibbs entropy is a measure of our uncertainty in the state of the system, which in this case are the locations of the diffusing particles. But we can choose the drift and diffusion coefficients such that the final distribution is narrower than the initial one

---

[2] $\rho_t^{\text{eq}} \equiv \rho^{\text{eq}}(\cdot, \lambda_t)$ depends on time only through the parameter $\lambda_t$.

[3] We are specifically referring to systems with a large number of degrees of freedom, $N$; if $N$ is small enough it is possible to obtain 'second law violating' behavior, see for instance [49, 67].

[4] Eq. (72) is also called the *differential entropy* of a continuous random variable, which has some important differences from the discrete version. Refer to chapter 8 of [70] for details.

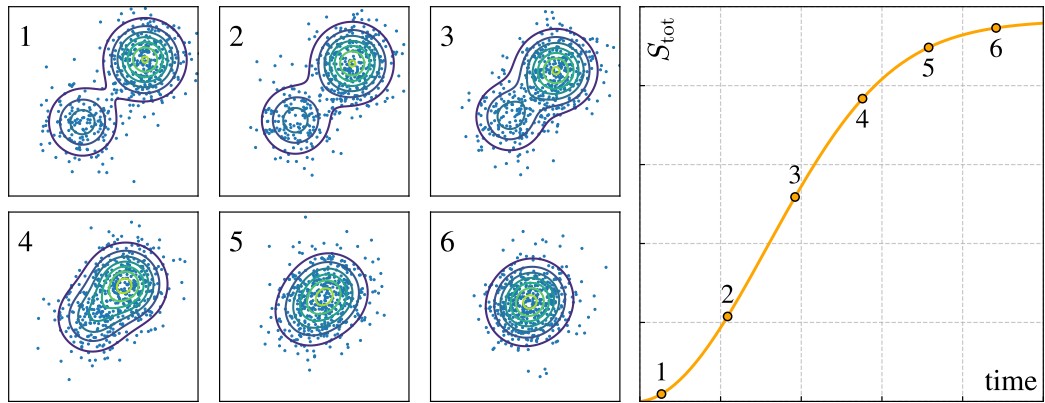

Figure 7: Diffusion is a non-equilibrium process that generates entropy over time. On the left, we see snapshots of a diffusive process (Ornstein-Uhlenbeck). In the forward direction (increasing $s$) the distribution evolves from $1 \to 6$ (i.e. $p_\mathrm{d} \to p_0$), and entropy produced till that point in time is indicated on the right. Note that $S_\mathrm{tot}$ is the total entropy produced, which is different from the change in Gibbs entropy of the distribution, which is negative in this experiment.

(see Fig. 7). Then, our ignorance of the particle positions would be reduced, so the change in Gibbs entropy is *negative*. However, the total entropy increases, as expected; $S_\mathrm{tot}$ is not just the change in Gibbs entropy.

We can see this explicitly by looking at the expressions for both. Combining Eqs. (66) and (72) and integrating by parts one obtains

$$S_\mathrm{G}[p_0] - S_\mathrm{G}[p_\mathrm{d}] = \int_0^T \mathrm{d}s\, \mathbb{E}_{\overleftarrow{p}} \left[ \frac{\sigma^2}{2} \left\| \nabla \log \overleftarrow{p} \right\|^2 + \nabla \cdot b_+ \right], \tag{73}$$

which is different from the total entropy,

$$S_\mathrm{tot} = \int_0^T \mathrm{d}s\, \frac{\sigma^2}{2} \mathbb{E}_{\overleftarrow{p}} \left[ \left\| \frac{2b_+}{\sigma^2} - \nabla \log \overleftarrow{p} \right\|^2 \right] \overset{(70)}{=} \mathrm{D}_{KL}\left(p_\mathrm{d}\big\|p_\mathrm{eq}^{(T)}\right) - \mathrm{D}_{KL}\left(p_0\big\|p_\mathrm{eq}^{(0)}\right). \tag{74}$$

These expressions differ when $b_+ \neq 0$. To understand how they are related we look at the free energy

$$F[p] = E[p] - \beta^{-1} S_\mathrm{G}[p], \tag{75}$$

where the temperature $\beta^{-1} := \sigma^2/2$ and the energy is

$$E[p] := \mathbb{E}_p\left[U(x)\right], \qquad U(x) = -\int^x \mathrm{d}\bar{x}\, b_+(\bar{x}). \tag{76}$$

For simplicity we will assume that $b_+$ and $\sigma$ are time-independent (cf. Eq. (5)), so we have a static equilibrium state $p_\mathrm{eq} = Z^{-1} \exp(-\beta U(x))$, and that $p_0 = p_\mathrm{eq}$. The generalization to the time-dependent case is straightforward. Then,

$$S_\mathrm{tot} = \mathrm{D}_{KL}\left(p_\mathrm{d}\big\|p_\mathrm{eq}\right) = -S_\mathrm{G}[p_\mathrm{d}] + \beta E[p_\mathrm{d}] + \log Z = \beta(F[p_\mathrm{d}] - F[p_\mathrm{eq}]), \tag{77}$$

where in the last step we have used $\beta F[p_\mathrm{eq}] = -\log Z$, which follows from evaluating Eq. (75) on $p_\mathrm{eq}$. Since $S_\mathrm{tot}$ is positive, $F[p_\mathrm{d}] > F[p_\mathrm{eq}]$ irrespective of whether $S_\mathrm{G}[p_\mathrm{d}]$ is larger or smaller than $S_\mathrm{G}[p_\mathrm{eq}]$. In Fig. 7, frame 1 has a higher Gibbs entropy but also a higher energy, so the particles coalesce into the distribution in frame 6, giving up some of their Gibbs entropy to move to a lower energy configuration. Thus, Langevin dynamics moves $p_\mathrm{d}$ towards the lower free energy state $p_\mathrm{eq}$ [71]. The minimization of free energy is also closely related to the interpretation of the Fokker-Planck equation as a Wasserstein gradient flow [72].

Relating $S_\mathrm{tot}$ to free energy also helps us connect the discussion in Sec. 2 to statistical mechanics. In systems with a large number of particles, the equilibrium distribution of microstates is sharply peaked, with the most probable microstates piled up around the free energy minima (see [3] or Sec. 4.6 in

[73]). The distributions $p$ are the microstates in the Schrödinger picture, and $\mathcal{P}[p] \propto \exp(-\beta N F[p])$ is peaked at $p_{\text{eq}}$. Then,

$$e^{-N S_{\text{tot}}} = \frac{e^{-\beta N F[p_{\text{d}}]}}{e^{-\beta N F[p_{\text{eq}}]}} \approx \mathcal{P}[p_{\text{d}}]. \tag{78}$$

Modifying the drift term in Eq. (9) changes the free energy landscape such that $p_{\text{d}}$ becomes its new minima. In other words, the diffusion model applies an external force to do work on the system, and the range of possible outcomes of the combined arrangement constitutes a non-equilibrium analog of the *Gibbs ensemble*.

## D   Score matching

In Sec. 4 we touched on the difficulty in defining neural entropy for the score-matching model. This point warrants further elaboration. We start with

$$\mathrm{d}X_t = -(b_+(X_t, T - t) - \sigma(t)^2 \nabla \log p(X_t, t))\mathrm{d}t + \sigma(T - t)\mathrm{d}B_t, \tag{79a}$$
$$\mathrm{d}X_t = -b_+(X_t, T - t)\mathrm{d}t + \sigma(T - t)\mathrm{d}B_t. \tag{79b}$$

Notice that the drift term in Eq. (79b) has the opposite sign to the one in Eq. (13b). This seems like a natural choice, since modifying $-b_+ \to -b_+ + \sigma^2 s_{\boldsymbol{\theta}}$ sets up the model to learn the score $\nabla \log p$ Eq. (19). But note that $b_+$ is a confining drift term which means $-b_+$ is not. Therefore, Eq. (79b) does not have a quasi-invariant distribution, and the intuition from Sec. 4 no longer holds. We can identify the inconsistencies arising from this conceptual breakdown through a straightforward calculation.

Starting with Eq. (11), let us tentatively define the ideal score matching neural entropy

$$\hat{S}_{\text{NN}}^{\text{sm}}(T) := \int_0^T \mathrm{d}s \, \frac{\sigma^2}{2} \mathbb{E}_p \left[ \|\nabla \log p\|^2 \right]. \tag{80}$$

The quantity on the right can be related to the non-equilibrium Gibbs entropy of the diffusing distribution, which we will write in terms of the time variable $s$ (cf. Eq. (72)). The change in Gibbs entropy under the forward process is given in Eq. (73). We may therefore rewrite Eq. (80) as

$$\hat{S}_{\text{NN}}^{\text{sm}}(T) = S_{\text{G}}[p_0] - S_{\text{G}}[p_{\text{d}}] - \int_0^T \mathrm{d}s \, \mathbb{E}_p \left[ \nabla \cdot b_+ \right]. \tag{81}$$

To simplify further we need to choose the drift term $b_+$. Let us consider the Variance Preserving (VP) process [1, 6], for which $b_+(y, s) = -\beta(s)y/2$ and $\sigma(s) = \sqrt{\beta(s)}$ in Eq. (40), where $\beta(s)$ is positive. Noting that $\nabla \cdot b_+ = -\frac{1}{2}\beta(s)\nabla \cdot x = -\frac{D}{2}\beta(s)$, Eq. (81) reduces to

$$\hat{S}_{\text{NN}}^{\text{VP}}(T) = S_{\text{G}}[p_0] - S_{\text{G}}[p_{\text{d}}] + \frac{D}{2} \int_0^T \mathrm{d}s \, \beta(s). \tag{82}$$

Upon closer inspection, Eq. (82) reveals a problem with the score matching neural entropy. Consider the trivial case where $p_0$ and $p_{\text{d}}$ are identical. The Gibbs entropies at the initial and final times are equal, therefore the first two terms cancel. But we are still left with a positive integral on the r.h.s. (see Fig. 8b). This is also apparent from Eq. (80)—the score function is static but it is not zero, so the neural entropy is some positive number, and 'information' is delivered to the network. Next, imagine changing $p_{\text{d}}$ to some other distribution $p_{\text{d}}'$, with $p_0$ and the $\beta$-schedule kept fixed. The only change in Eq. (11) is the $-S_{\text{G}}[p_{\text{d}}]$ term. As a result, if $p_{\text{d}}'$ has a larger entropy than $p_{\text{d}}$ the neural entropy decreases. It would seem that the network needs to remember *less* information to transform $p_0 \to p_{\text{d}}'$ than it does to convert $p_{\text{d}}$ to itself!

The issue arises from the $-b_+$ term in Eq. (79). For the VP process that SDE is

$$\mathrm{d}X = \frac{\beta}{2} X_t \mathrm{d}t + \sqrt{\beta}\mathrm{d}B_t. \tag{83}$$

But this process has a repulsive drift term which, given enough time, dilutes the distribution away to infinity. Therefore, the network $s_{\boldsymbol{\theta}}$ has to *work against* $-b_+$ to keep the distribution $p_{\boldsymbol{\theta}}$ intact. In the above examples the growth in neural entropy due to the $\int \mathrm{d}s\beta(s)$ term in Eq. (82) is indicative of the effort needed to hold the diffusing particles in place as the drift $\beta X/2$ tries to drive them apart (see

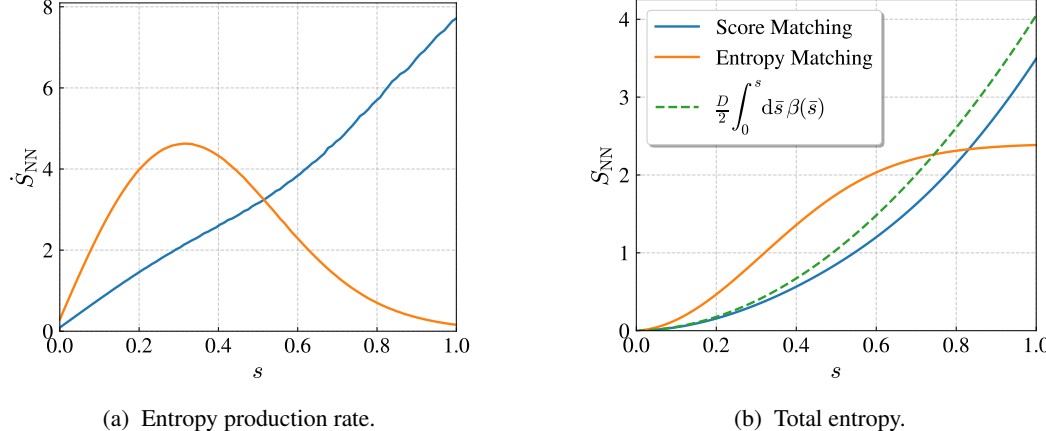

(a) Entropy production rate.

(b) Total entropy.

Figure 8: Ideal neural entropy curves for score matching and entropy matching models with a VP process. For entropy-matching models, the neural entropy is the same as $S_{\text{tot}}$ (cf. Eq. (11)), which is why entropy production trails off as forward diffusion approaches its final state.

Fig. 8). For this reason, the score matching neural entropy from Eq. (80) is not an accurate gauge of the non-trivial information the network must store to reverse diffusion.

In practice, Eq. (80) can be approximated by

$$S_{\text{NN}}^{\text{sm}}(T) := \int_0^T \mathrm{d}s \, \frac{\sigma^2}{2} \mathbb{E}_p \left[ \|\boldsymbol{s_\theta}\|^2 \right], \tag{84}$$

just like we did in Eq. (18). Experimental comparison of the neural entropy in entropy-matching models and the $S_{\text{NN}}^{\text{sm}}$ for score-matching models are shown in Fig. 9.

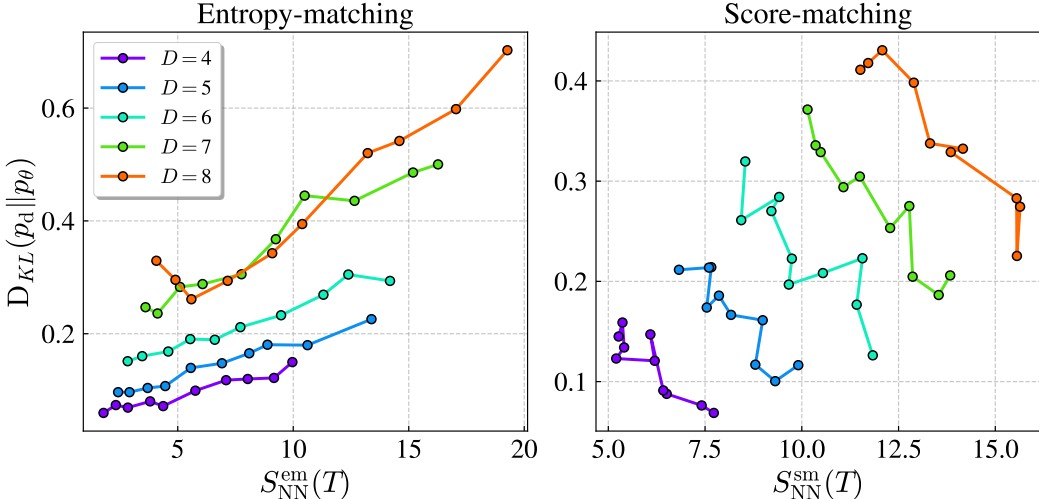

Figure 9: KL versus neural entropy for the entropy-matching and score-matching models. Both models are trained on a series of progressively larger Gaussian mixtures, just like the ones used for Fig. 11. The experiments are repeated in different dimensions, $D$. $S_{\text{NN}}^{\text{em}}$ is the neural entropy defined in Eq. (11). $S_{\text{NN}}^{\text{sm}}$ is an analogous, but different, quantity defined in App. D in an attempt to define neural entropy for a score-matching model. Note how network performance *decreases* at lower values of $S_{\text{NN}}^{\text{sm}}$, in contrast to the trend in entropy-matching. This behavior is explained in App. D. The VP process was used for both sets of experiments, with the same $\beta$-schedule.

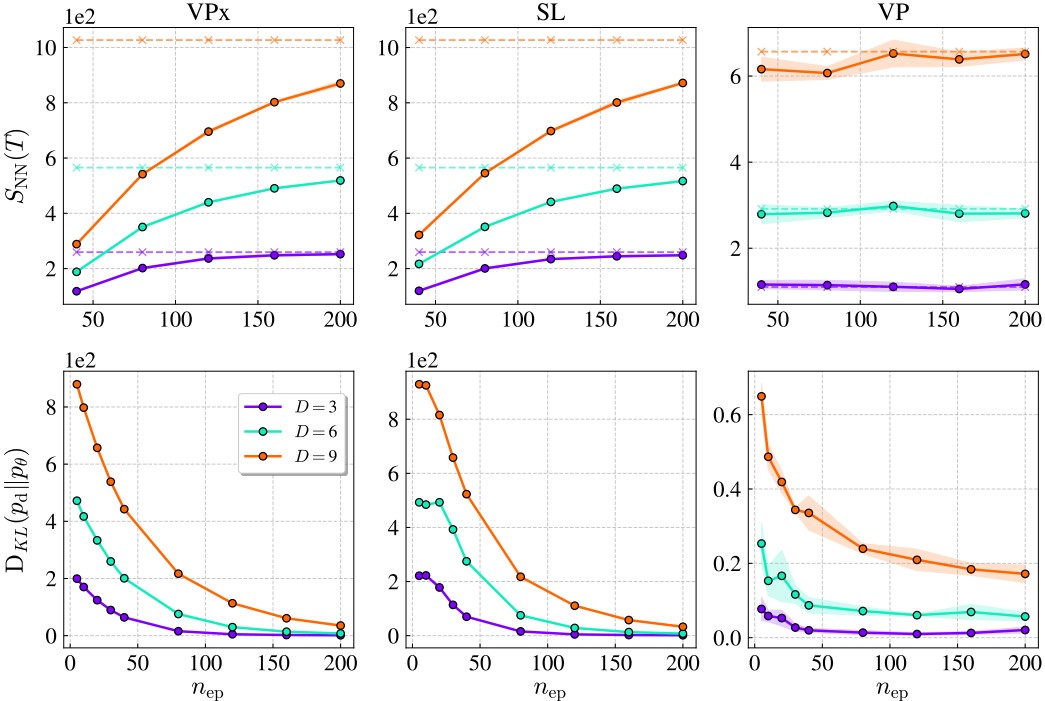

Figure 10: The evolution of neural entropy (top) and the KL between $p_\mathrm{d}$ and the reconstructed distribution $p_{\boldsymbol{\theta}}(\cdot, T)$ (bottom) over training epochs, $n_\mathrm{ep}$. The entropies and KL are measured at $s = T = 1$. The $p_\mathrm{d}$'s are Gaussian mixtures in $D = 3, 6, 9$ dimensions. The dashed lines are the actual value of $S_\mathrm{tot}$ generated by the respective diffusion processes. The VPx and SL processes ($\kappa = \sigma_0 = 0.1$) produce two orders of magnitude larger $S_\mathrm{tot}$ than VP, which is why $S_\mathrm{NN}$ is slow to catch up in these models—the network takes longer to absorb more information. This increase in retention is tracked by a decrease in the KL. Here again, the VP model outshines VPx and SL: the diffusion model can reconstitute $p_\mathrm{d}$ more faithfully when it has to remember less information to do so.

# E  Details of experiments

All models in this paper, except for the ones in Fig. 9, were trained with 4 random seeds varying both weights initialization and order of training data, and the results were averaged over. The faint bands in the plots are within one standard deviation from the mean measurements. All computations were done on A100 GPUs with 80 GB of memory. The CIFAR-10 models were trained on 4 GPUs in parallel while the Gaussian mixture and MLP experiments were trained on just one. Training on CIFAR-10 with the full dataset ($n_c = 5000$ in Fig. 16) takes 4.5 hours. Experiments for MNIST that stop between training epochs to compute log densities (see Figs. 3 and 13), and repeat for different numbers of training samples, take around 4 hours for each training seed. For the low-$D$ models in the transport experiments we used Fourier features on the $x$ variable to help the the MLPs learn better [74]. These were inserted before the input stage of an MLP with architecture $(512, 256, D)$. We use $T = 1$ in all experiments. More details about specific experiments are given in the respective figure captions.

## E.1  Diffusion models

We use three kinds of diffusion processes to experiment with different entropy production profiles in diffusion models. These are the VP, VPx, and Straight Line (SL) SDEs from Eq. (24) and Eq. (20) respectively. These are Ornstein-Uhlenbeck processes, which have the general form [29]

$$\mathrm{d}Y_s = f(s)Y_s\mathrm{d}s + \sigma(s)\mathrm{d}B_s. \tag{85}$$

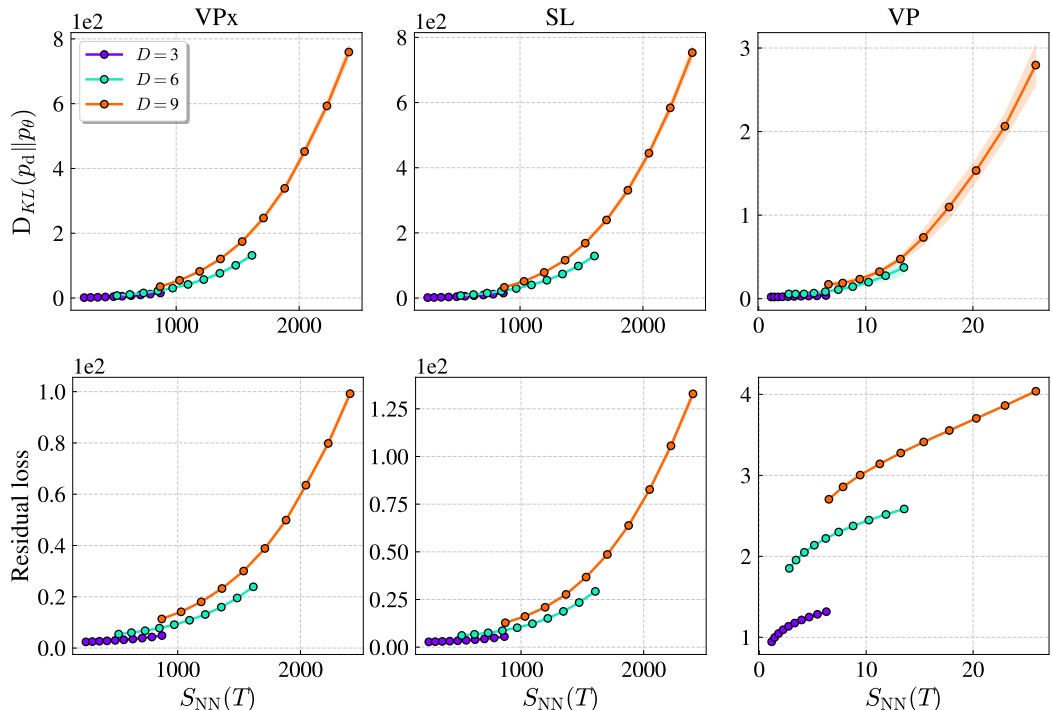

Figure 11: KL and residual loss at $n_{\text{ep}} = 200$ epochs vs. the neural entropy in the network. The plots are generated with the same experimental setup as Fig. 10, but we vary the total entropy produced by using increasingly broader $p_{\text{d}}$'s.

The perturbation kernel of this SDE is

$$p(y_s|y_0) = \mathcal{N}\left(y_s; \mu(s)y_0, \Sigma(s)^2 \mathbb{1}_D\right), \tag{86}$$

where

$$\mu(s) = \exp\left(\int_0^s \mathrm{d}\bar{s} f(\bar{s})\right), \tag{87a}$$

$$\Sigma(s)^2 = \mu(s)^2 \int_0^s \mathrm{d}\bar{s} \frac{\sigma(\bar{s})^2}{\mu(\bar{s})^2}. \tag{87b}$$

Starting at $s = 0$, a sample $y_{\text{d}} \sim p_{\text{d}}$ is propagated to

$$y_s = \mu(s)y_{\text{d}} + \Sigma(s)\epsilon \tag{88}$$

at an intermediate time $s \in (0, T]$, where $\epsilon \sim \mathcal{N}(0, \mathbb{1}_{\text{D}})$. The object

$$\nabla \log p(y_s|y_{\text{d}}) = -\frac{\epsilon}{\Sigma(s)} \tag{89}$$

is used in the *denoising* entropy-matching objective

$$\mathcal{L}_{\text{DEM}} := T \, \mathbb{E}_{y_{\text{d}} \sim p_{\text{d}}} \mathbb{E}_{s \sim \mathcal{U}(0,T)} \left[ \Lambda(s) \frac{\sigma(s)^2}{2} \mathbb{E}_{y_s \sim p(y_s|y_{\text{d}})} \left\| \nabla \log p_{\text{eq}}^{(s)} + \boldsymbol{e}_{\boldsymbol{\theta}} - \nabla \log p(y_s|y_{\text{d}}) \right\|^2 \right]. \tag{90}$$

It is straightforward to show that $\mathcal{L}_{\text{DEM}}$ is equivalent to the upper bound in Eq. (17) when $\Lambda(s) = 1$ [76]. For instance, plugging $\boldsymbol{s}_{\boldsymbol{\theta}} = \nabla \log p_{\text{eq}}^{(t)} + \boldsymbol{e}_{\boldsymbol{\theta}}$ into the derivation in App. A of [40] would suffice. Empirically, it has been found that the choice $\Lambda(s) = 2\Sigma(s)^2/\sigma(s)^2$ produces better results in image models [5], although [20] reports that $\Lambda(s) = 1$ gives better log densities when used in conjunction with importance sampling. We have used the prescription from [5] with $s$ sampled from the uniform distribution $\mathcal{U}(0, T)$ with no importance sampling. In the experiments shown in Figs. 2 and 10 each $y_{\text{d}}$ is evolved to 10 random values of $s$ from this interval, which improved KL estimates whilst also

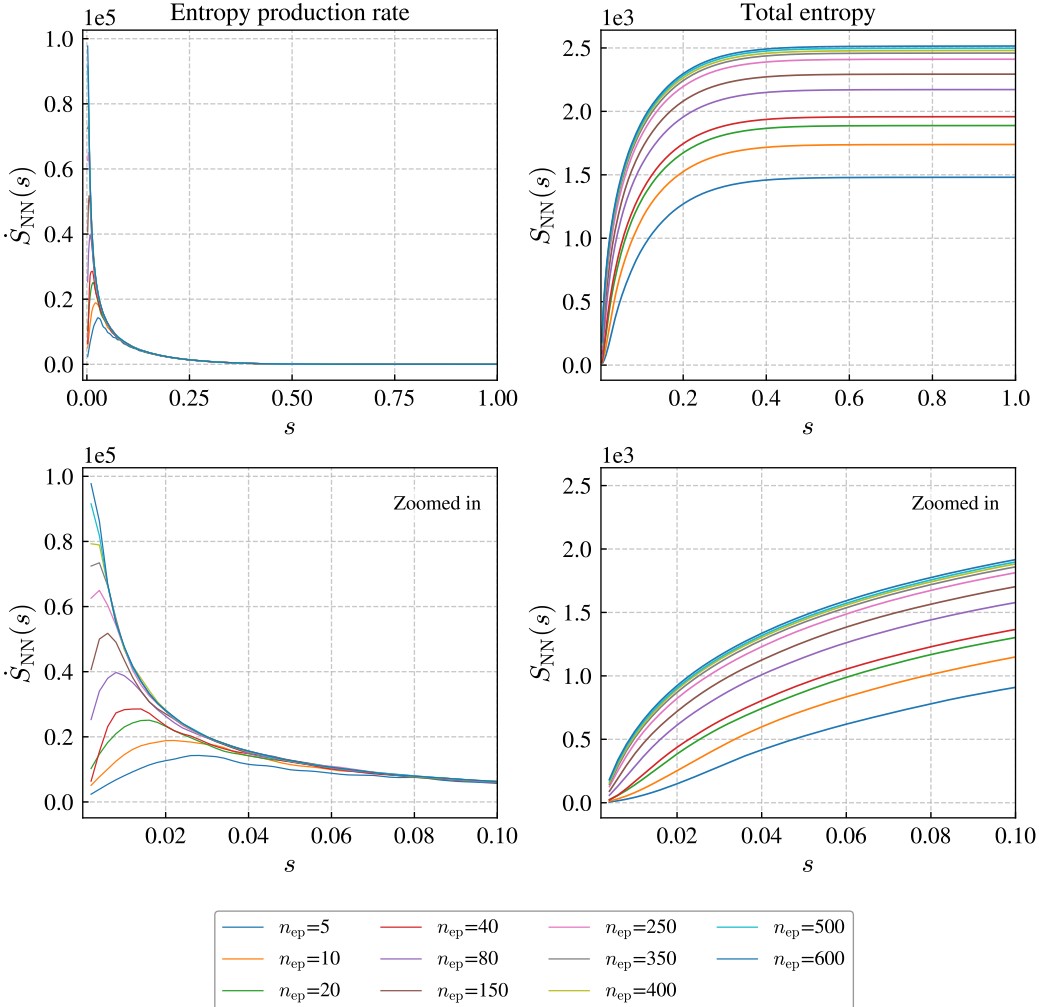

Figure 12: Entropy production curves for an image diffusion model trained on the entire MNIST dataset ($n_c = 6000$ from Fig. 3) with the VP process. The different lines correspond to entropy measurements at different epochs of training (see Fig. 2). The lower panels zoom in on a time interval close to the start of the forward diffusion process. Notice how the entropy production rate is sharply peaked near $s = 0$. This is due to the dimensionality of the data manifold $\mathcal{M}_{\mathrm{d}}$. Since $\mathcal{M}_{\mathrm{d}}$ is much lower dimensional than the ambient pixel space the model must supply a large amount of information as $t \to T$ (or $s \to 0$) to place the samples on $\mathcal{M}_{\mathrm{d}}$. The same behavior also appears in score-matching models, but it is often conflated with a numerical divergence at $s = 0$ due to the vanishing of $\Sigma(s)$ in Eq. (89) [75]. The latter is addressed by truncating the diffusion process at $s = 10^{-5}$ [6]. These entropy production rates are computed by splitting the interval $(10^{-5}, T]$ into 500 time steps and computing the average in Eq. (23) over 1000 samples from the test dataset propagated to each step.

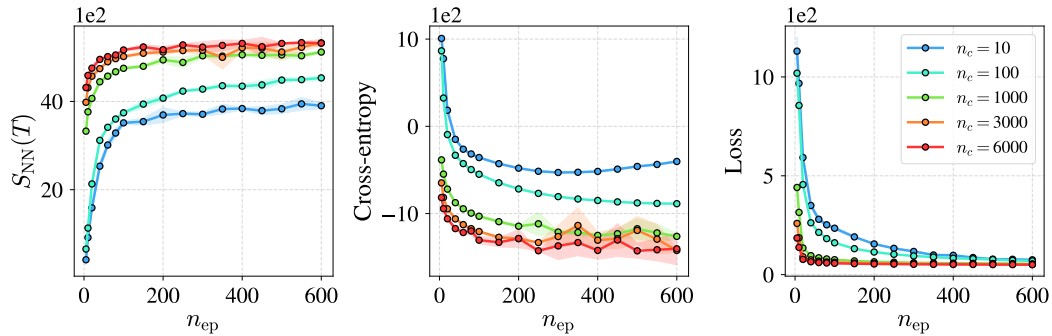

Figure 13: The evolution of neural entropy, cross-entropy, and loss over training epochs for an unconditional image diffusion model (SL) trained on the MNIST dataset. This is the analog of Fig. 3 for the Straight Line diffusion model (cf. Eq. (20)). Notice that a far greater amount of neural entropy is produced here compared to the VP process, for reasons explained in Sec. 5. However, the cross entropy settles to similar values for both VP and SL. The scaling of $S_{\mathrm{NN}}(T)$ with the number of samples, at $n_{\mathrm{ep}} = 600$, still exhibits a nearly logarithmic trend, just like with the VP process (see Fig. 16).

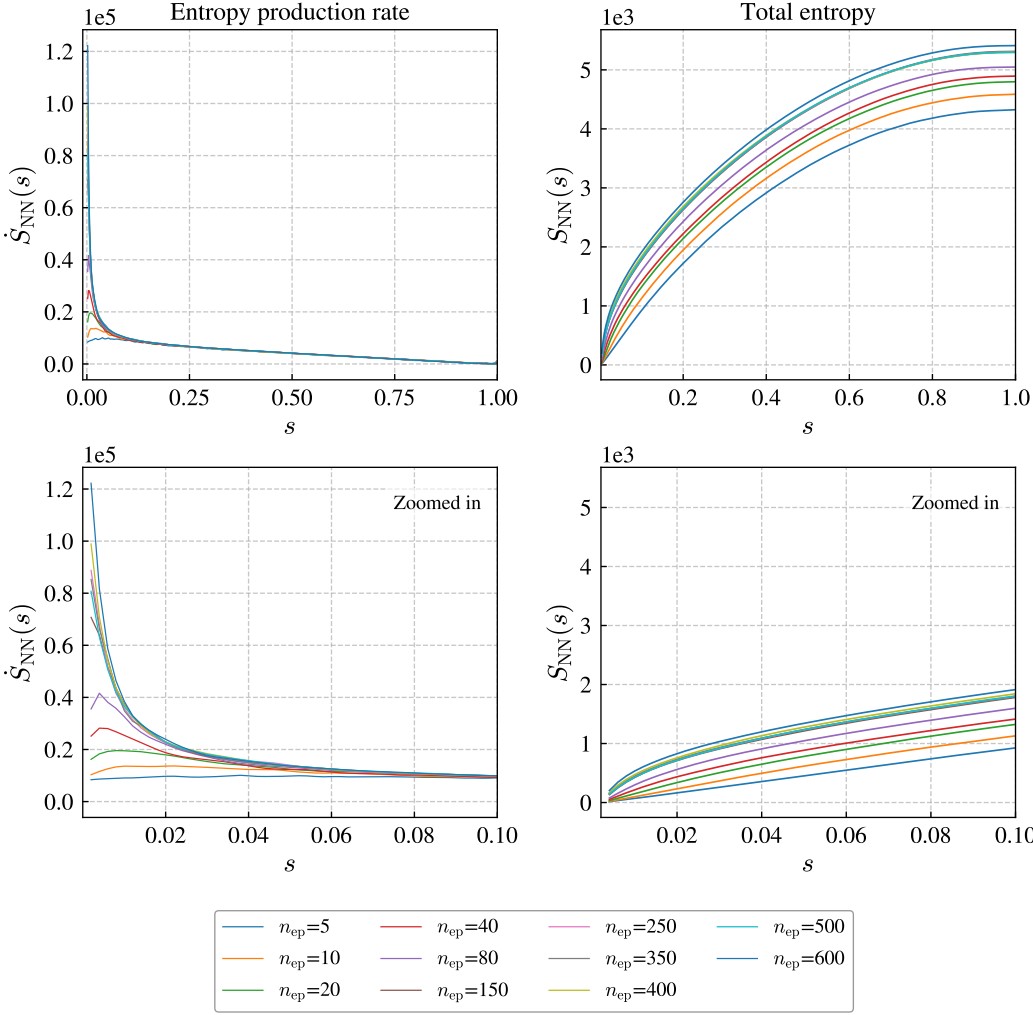

Figure 14: Entropy production curves for an image diffusion model trained on the entire MNIST dataset ($n_c = 6000$ from Fig. 13) with the SL process.

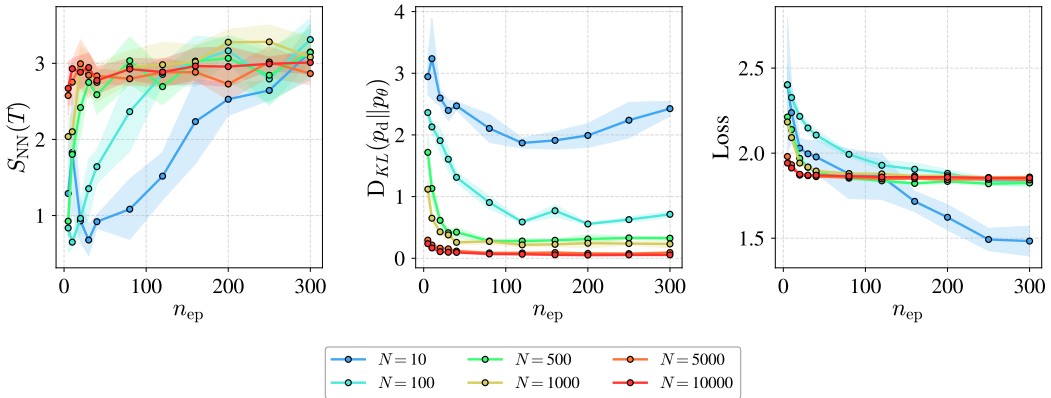

Figure 15: The evolution of neural entropy, cross-entropy, and loss over training epochs for diffusion model (VP) with an MLP core trained on a mixture of five Gaussians in $D = 6$ dimensions. $N$ is the number of samples used for training. The scaling of $S_{\mathrm{NN}}(T)$ with the number of samples does *not* show a neat trend like the ones for the image models (see Fig. 16). Due to the unstructured nature of the data and the relatively weak prior constraints imposed by the MLP, the model learns different distributions at different $N$. This is most emphatic for $N = 10$ where the data is so sparse that the model increasingly concentrates probability mass around the given samples as training progresses. This is why the KL rises and loss continues to drop for $N = 10$.

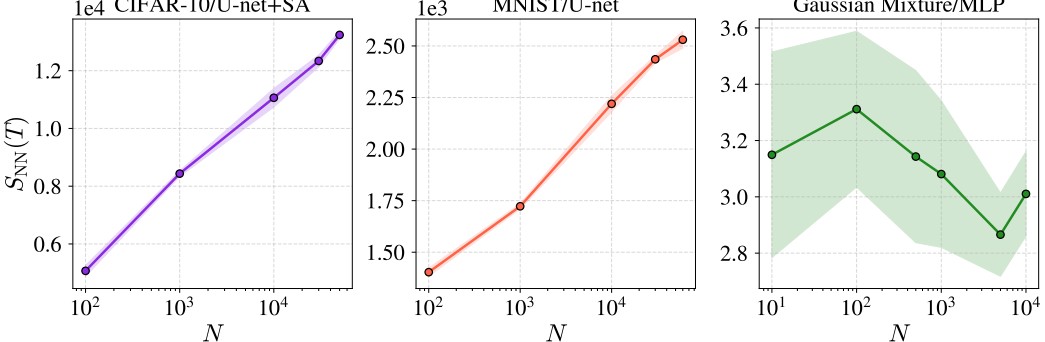

Figure 16: Neural entropy versus number of samples for CIFAR-10 trained on a U-net with self-attention layers (left), MNIST trained on a simple U-net (center) (cf. Fig. 3), and mixture of Gaussians in $D = 6$ trained on an MLP-based diffusion model (right) (cf. Fig. 15). These are the values of $S_{\mathrm{NN}}(T)$ at the end of training. The first two plots are the same ones from Fig. 1. Note the absence of the logarithmic trend in the Gaussian mixture/MLP case. All models shown here use the VP process.

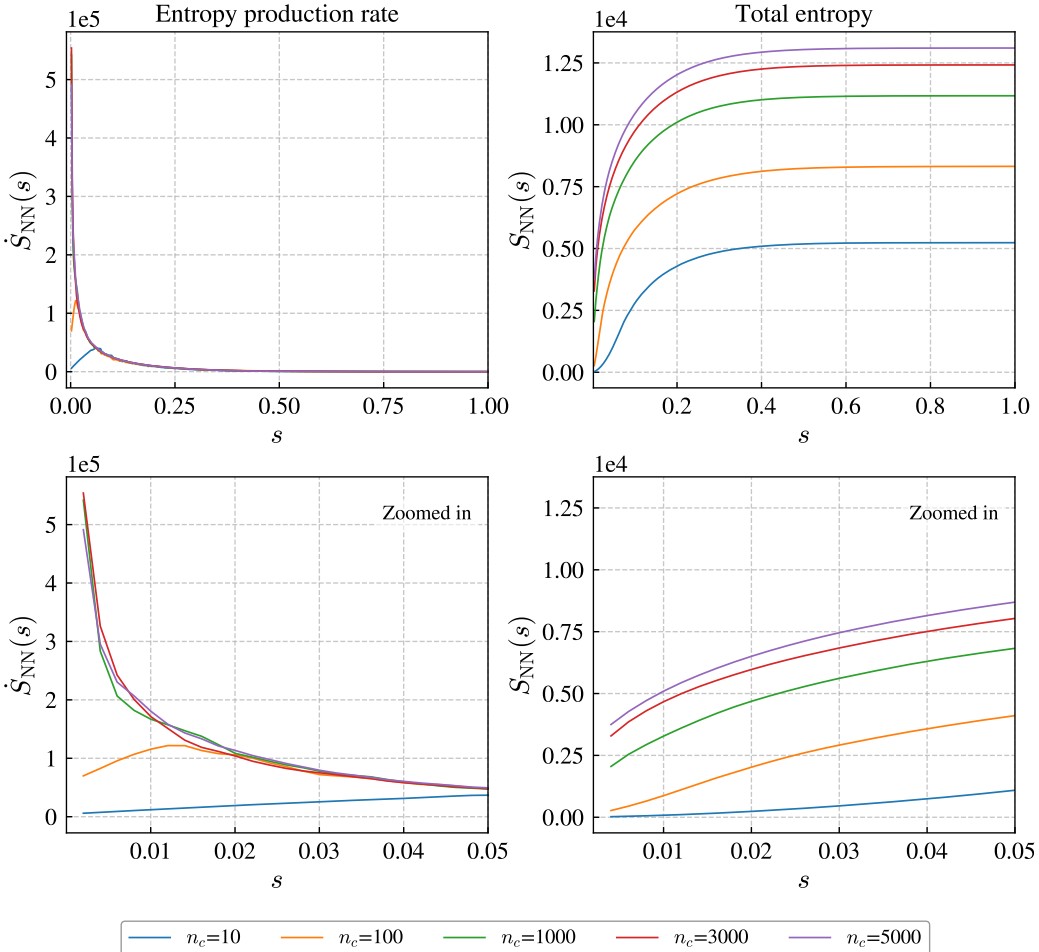

Figure 17: Entropy production curves for CIFAR-10. Note that the different colors correspond to the different number of samples per class used for training, $n_c$, rather than the number of epochs. All CIFAR-10 experiments were trained to 200 epochs. Computing the neural entropy in this case requires special care since the peak near $s = 0$ is even sharper than the ones for MNIST (see Fig. 12); the lower-dimensional data manifold with the CIFAR-10 images lives in a much higher-dimensional pixel space compared to MNIST.

reducing loss fluctuations [77, 78]. But in the image models we sampled at just one random $s$ per $y_d$ per epoch.

The functions $\mu(s)$ and $\Sigma(s)$ for the VPx and SL processes can be read off from Eq. (25) and Eq. (21) respectively. In both cases $\Sigma(s)$ vanishes at $s = 0$, so Eq. (89) diverges at that instant. Therefore we do not venture below $s = 10^{-5}$ when training with Eq. (90). The SL SDE has an additional singularity at $s = T = 1$, so we also cut off samples at $s = 1 - 10^{-5}$ in that case. Note that [31] approximates $\Sigma(s)_{\text{SL}} \approx \sigma_0$ since $\sigma_0$ is small, but we retain the full time-dependence in our experiments.

New samples from a diffusion model can be generated efficiently using the Probability Flow ODE [6, 79, 80],

$$\mathrm{d}x(t) = -\left( b_+(x, T - t) - \frac{\sigma^2}{2} \nabla \log p(x, t) \right) \mathrm{d}t, \tag{91}$$

where the true score is approximated by $\nabla \log p \approx \nabla \log p_{\text{eq}}^{(t)} + e_{\boldsymbol{\theta}}$ in entropy-matching models.

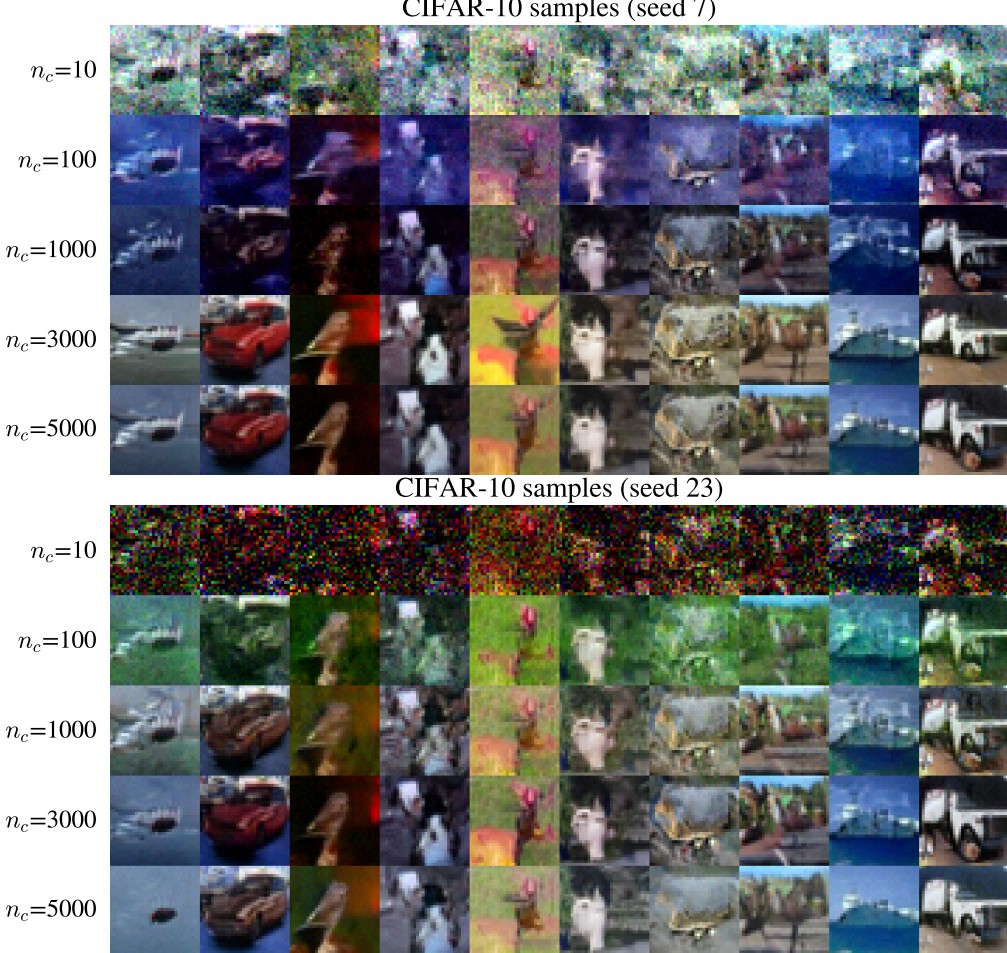

Figure 18: A few samples generated from the conditional diffusion model trained on CIFAR-10 images. Each row in a grid contains one image from each class, and the rows correspond to models trained with different numbers of samples $n_c$ per class. We used the probability flow ODE to produce these images (cf. Eq. (91)), with the *same* ten initial noise vectors for each $n_c$. These samples affirm the key takeaway from the neural entropy vs. $N$ trends Figs. 1 and 16: the rate of additional information absorbed by the model decreases with each new sample. The two grids differ by the seed value used to initialize model weights and fix the order in which the training samples are applied.

## E.2 Density estimation

In our experiments with Gaussian mixtures and MNIST, we compute the KL divergence to the true distribution and the cross-entropy respectively to gauge mode performance (cf. Figs. 3, 13 and 15). But diffusion models do *not* give exact log densities on the learned distribution, despite claims in the literature. However, a lower bound on the log density $\log p_{\boldsymbol{\theta}}(x, T)$ can be established from Eq. (61). That latter is, explicitly,

$$
\log p_u(x, T) \geq -\mathbb{E}\left[\int_0^T \mathrm{d}s \left(\frac{1}{2\sigma^2}\|b_+ - u\|^2 - \nabla \cdot u\right) - \log p_u(Y_T, 0)\Big|Y_0 = x\right].
\tag{92}
$$

The expectation is computed over trajectories generated by Eq. (40) that start at $x$ at $s = 0$. The bound is saturated if $u = b_-$, in which case $p_u(x, T) = p_{\mathrm{d}}(x)$ (cf. App. B.4), but $u$ is approximated by a neural network in diffusion models. We can use integration by parts to avoid taking the gradient

of the neural network in the $\nabla \cdot u$ term. For a vector-valued function $h(y_s, s)$,

$$
\begin{aligned}
\mathbb{E}_{Y_s}\left[\nabla \cdot h(Y_s, s) | Y_0 = x\right] &= -\int \mathrm{d}y_s \, h(y_s, s) \cdot \nabla p(y_s, s | x, 0) \\
&= -\mathbb{E}_{Y_s}\left[h(Y_s, s) \cdot \nabla \log p(Y_s, s | Y_0, 0) | Y_0 = x\right],
\end{aligned}
\tag{93}
$$

where $p(y_s, s | x, 0)$ is the same kernel from Eq. (89) with the time-dependence indicated explicitly, and we have assumed that the product $hp$ vanishes at the $x$-boundaries. Using Eq. (93) in Eq. (92) we obtain

$$
\log p_u(x, T) \geq
\tag{94}
$$
$$
- \mathbb{E}\left[\int_0^T \mathrm{d}s \left(\frac{1}{2\sigma^2} \|b_+ - u\|^2 + u \cdot \nabla \log p(Y_s, s | Y_0, 0)\right) - \log p_u(Y_T, 0) \Big| Y_0 = x\right].
$$

By transferring the gradient operator from $u$ we avoid the need to take derivatives of the neural network; since the transition probability is a Gaussian the gradients of their log are easy to calculate. The r.h.s. is a path integral, which can estimate efficiently as a Monte Carlo average

$$
\log p_u(x, T) \geq -T \, \mathbb{E}_{s \sim \mathcal{U}(0,T)} \mathbb{E}_{y_s \sim p(y_s, s | x, 0)}\left[\frac{1}{2\sigma^2} \|b_+ - u\|^2 + u \cdot \nabla \log p(y_s, s | x, 0)\right] - S_{\mathrm{G}}[p_0].
\tag{95}
$$

Here, we have replaced $\mathbb{E}_{y_T \sim p(y_T, T | x, 0)}\left[\log p_u(y_T)\right]$ with the negative Gibbs entropy $-S_{\mathrm{G}}[p_0]$ since $y_T$ would be distributed as $p_0$ irrespective of the $x$ at which it started, to a very good approximation (cf. App. C.2). Finally, for entropy-matching models, $u = -b_+ - \sigma^2 e_{\boldsymbol{\theta}}$, and therefore

$$
\log p_{\boldsymbol{\theta}}^{\mathrm{em}}(x, T) \geq - S_{\mathrm{G}}[p_0]
$$
$$
-T \, \mathbb{E}_{s \sim \mathcal{U}(0,T)} \mathbb{E}_{y_s \sim p(y_s, s | x, 0)}\left[\frac{\sigma^2}{2}\left\|\nabla \log p_{\mathrm{eq}}^{(t)} + e_{\boldsymbol{\theta}}\right\|^2 - (b_+ + \sigma^2 e_{\boldsymbol{\theta}}) \cdot \nabla \log p(y_s, s | x, 0)\right].
\tag{96}
$$

We use this lower bound in lieu of the true neural log densities in our calculations. The MC average must be computed with a fairly large number of samples [77, 78]. For the Gaussian mixture experiments, this is an excellent substitution. The results are noisy but still insightful in the image experiments. A version of Eq. (96) for score-matching is derived in [20], which is obtained by setting $e_{\boldsymbol{\theta}} = -\nabla \log p_{\mathrm{eq}}^{(t)} + s_{\boldsymbol{\theta}}$ in this one.

