# OpenReview forum: "Neural Entropy"
_NeurIPS.cc/2025/Conference — NeurIPS 2025 spotlight_

### Official Review · Reviewer_mGKE · 2025-06-23

**Clarity:** 4
**Significance:** 3
**Originality:** 4
**Rating:** 5
**Confidence:** 4

**Summary:**

This work introduces the concept of neural entropy, to quantify the amount of information stored in a neural network. The focus of this work is on diffusion-based generative models, and thus the “score network” or “denoising network” that is typically used to learn how to reverse the diffusion dynamics. Throughout a very didactical derivation, the authors argue that diffusion models are very effective in storing information, especially when dealing with large amounts of structured data: this claim is supported by mathematical derivations as well as simple experiments both on synthetic data and on simple image datasets.

Using neural entropy, the authors show that the growth of neural entropy with the number of training samples in diffusion models obeys to logarithmic scaling, suggesting that the marginal information gained per sample decreases as approximately $1/N$. Moreover, they also show that there is some sort of “speed limit” in the sense that diffusion models that employ “tricks” to make them faster, also need to store more information for an accurate reversal process.

Furthermore, this work introduces the entropy-matching objective, as a way to tighten the KL divergence between the data distribution and its parametric approximation achieved with a “denoising network”. This is in contrast to a score-matching objective, for which a neural entropy expression is conceivable, but hard to interpret, especially in the case of a variance preserving process. Instead, the entropy-matching objective has a clearer interpretation in terms of neural entropy.

Finally, this article presents a series of simple experiments to validate numerically the main theory of the paper: empirical lines match the predictions of the ideal ones, supporting the claims of the authors.

**Questions:**

* What are the main practical implications of this work? Can you illustrate how your ideas can help designing new diffusion-based generative models? This kind of discussion could help broaden the scope and audience of your work

* I think the problems with score-based diffusion models, having a drift term that is “repulsive”, is important. The discussion in the appendix and in particular Figure 9, right-hand side, points at a possible mismatch of the neural entropy metric for score-based models. Can you help me clarify wether this implies that the proposed method to measure information stored in a score-matching based model is limited, or if there are any important limitations to score-matching models due to this drift term?

* Recently, there has been a significant increase of work that use diffusion models, as well as other generative models to compute information theoretic quantities such as Entropy and Mutual Information. In those cases, the key objective is to compute the entropy of a given data distribution, using score networks for example, or to compute the mutual information between two data distributions, again using score networks for example. Do you think these works are relevant to some extent and could be discussed as related work?

[1] Butakov et. al., “Mutual Information Estimation via Normalizing Flows”, https://arxiv.org/abs/2403.02187

[2] Franzese et. al, “Mutual Information Neural Diffusion Estimation”, https://arxiv.org/abs/2310.09031

[3] Kholkin et. al, “InfoBridge: Mutual Information estimation via Bridge Matching”, https://arxiv.org/abs/2502.01383

**Ethical Concerns:**

["NO or VERY MINOR ethics concerns only"]

**Final Justification:**

I am happy with the rebuttal and the answers to my questions.

I really liked this work and believe the narrative rooted in physics is a nice contribution to the community, which will spark interest and discussions.

**Limitations:**

Yes.

**Paper Formatting Concerns:**

Did not notice formatting issues.

**Quality:**

3

**Strengths And Weaknesses:**

* Strengths:
  * This is a very easy to read article, very pedagogic and clear. It has many examples, and begins with simple processes inspired by established results in physics, but continue with more elaborate and realistic stochastic processes that are used in modern generative models.
  * This work presents a very intuitive explanation for the amount of information stored into a network used to reverse the dynamics of diffusion processes. The relation between information storage and the structure of the data, the number of training samples, and the particular form of the stochastic process used is important and could be used to better engineer diffusion-based generative models
  * The authors introduce the entropy-matching objective, which is easy to implement and optimize, similarly to flow-matching objectives, although care needs to be taken to select appropriate measures for numerical stability
  * I liked the discussion and perspectives in the conclusion, aiming at answering the question about the application of the proposed neural entropy measure to other generative models, such as language models. The connections to diffusion-based LLMs is pertinent and interesting

* Weaknesses:
  * I’m lucky enough to have sufficient background on the literature required to understand this paper, but for a generic reader, this work might be somehow difficult to digest, as it is not self-contained. The (long) appendix does a good job in providing background material and proofs, but it requires a particular effort to “put all pieces together”
  * The practical application of neural entropy to image diffusion models is hindered by some instabilities and calls for more sophisticated diffusion-based methods to avoid divergence
  * The authors did not put a particular effort in translating the methodological results presented in this work into actionable information for practitioners who might use entropy-matching, and neural entropy to design novel (and better) image diffusion models

---

> ### Author Rebuttal · Authors · 2025-07-26
>
> Thank you for the thoughtful review and the many constructive comments. We will try to address your concerns below.
>
> Weaknesses:
>
> 1. We acknowledge that not all readers in the ML community may have the necessary background in statistical physics. We have references several review articles and pedagogic papers throughout this work to direct readers toward the necessary preliminaries. Rather than repeating these materials with reduced clarity or completeness, we opted to focus on presenting the novel contributions of this work, which already makes for a lengthy paper, as you noted. We have assumed that basics of information theory, at the level of earlier chapters in McKay or Cover and Thomas, is familiar to the reader. If there are resources that present this material in a more accessible way to NeurIPS readers we are happy to cite them in the paper.
>
> 2. If the divergence at initial time, $s=0$ is what is referred to as the instability in entropy matching, we should point out that the same phenomenon appears in score-matching models. There is some discussion of this in the caption for Figure 12. We point out that, for image models, there are two divergences sitting on top of one another at this instant. The first is due to the vanishing of $\Sigma(s)$ in $\nabla \log p(y_s|y_0) = - \epsilon/\Sigma(s)$. But the second is due to the fact that the image manifold is much lower dimensional than the ambient pixel space. Score-matching suffers from both divergences as well.
>
> 3. We have follow up papers in preparation that address precisely this. This paper is meant to be the first of a series. Our goal here was to introduce the concept of neural entropy and justify that it does indeed measure the information stored in the network. That said, there is a comparison between the SLDM and VP processes on MNIST images in Fig. 12 and 14.
>
> Questions:
>
> 1. The entropy picture allows us to gauge and compare the information load on the network under different choices of diffusion processes. For example, we mention that these results can be generalized to discrete diffusion, and therefore diffusion LLMs. In that case the thermodynamic speed limit can help us choose more optimal 'noising' processes to be applied on the training data. We plan to do some of these experiments ourselves. For image models, a relevant question is the neural entropy in latent space models. We suspect that the neural entropy for a model trained on the latents will be much smaller that one trained directly on the images.
>
> 2. Great question! The fundamental problem with defining neural entropy in score-matching models is that the repulsive drift term does not allow for a quasi-invariant state. In practice, this means that a score-matching network has to put additional effort into 'fighting' the repulsive drift term, to keep the random walkers from moving to infinity, while also reshaping the distribution back to $p_{\rm d}$. Although not shown in App. D, we found that the entropy matching models train faster (that is, the loss converges quicker) than score matching models in experiments involving Gaussian models. This makes sense, since entropy-matching puts less information 'strain' on the neural network than score-matching. These experiments can be found in another paper of ours, which is cited in the present work. We refrain from citing it here to preserve anonymity.
>
> 3. Thank you for the references. Indeed, there is a close relationship between neural entropy and mutual information. We are writing a paper at the moment which discusses this in greater detail, and we will make sure to discuss your references as related work. In addition, Sec. 3.2. of Franzese et. al is closely related to the entropy matching model in our paper, so we will cite this work in future revisions.
>
> Once again, we sincerely appreciate your comprehensive review, and your time.

---

> > ### Comment · Reviewer_mGKE · 2025-08-01
> > **Thank you for the rebuttal**
> >
> > Dear authors,
> > thank you for the response to my questions and for the discussions.
> >
> > I think your work is a valuable contribution to the community and that the "statistical physics perspective" is well motivated and solid. Answers to my questions are convincing, I will update my score.

---

> > > ### Author Response · Authors · 2025-08-01
> > > **Thank you**
> > >
> > > We appreciate your words of encouragement, and the revised score.

---

### Official Review · Reviewer_nu8R · 2025-06-24

**Clarity:** 3
**Significance:** 3
**Originality:** 3
**Rating:** 5
**Confidence:** 3

**Summary:**

The paper defines and analyses a measure called neural entropy. The definition of neural entropy is justified via concepts of information theory and thermodynamics applied to random walk processes and related to diffusion models in machine learning. The analysis of this measure is performed empirically in both a toy controlled setup, where the ground truth entropy is known, and an uncontrolled setup, using two common datasets in the machine learning community.

**Questions:**

## Questions

1. Section 6, paragraph “Transport experiments”: Good point to limit the inductive biases here, but what about the other effects on the neural entropy such as the size of the network, the choice of the optimisation algorithm, the batch size, the number of samples per epoch (or did you resample your Gaussians at each epoch as if an infinite amount of data was available?), etc.? Did you notice any interesting behaviour or was the neural entropy stable with respect to such hyperparameters? Sorry if I missed this in the text.

1. Did you try your method on a larger and much more complicated dataset such as ImageNet? As I acknowledge that one could always try more, it is not a weakness strictly speaking. Nevertheless, it would be interesting to check whether, at a certain point, there is a limit to the inductive biases of the network which would manifest via the neural entropy.

1. Section 6, paragraph “Storage experiments”: Do you have any quantitative argument to justify that the model is indeed not memorising?

## Suggestions

- Section 6, paragraph “Transport experiments”, line 4: There may be a typo with the reference to Eq. (15), which I think should be Eq. (17).

**Ethical Concerns:**

["NO or VERY MINOR ethics concerns only"]

**Final Justification:**

The response given by the authors, complemented by the rebuttal to reviewer KtyK, are convincing. The rating will remain the same.

**Limitations:**

Yes.

**Paper Formatting Concerns:**

No major issue.

**Quality:**

3

**Strengths And Weaknesses:**

## Strengths

- The paper clearly states, at multiple places, the limitations of the neural entropy measure and the experiments performed.

- The analysis performed gives several important insights on neural networks, namely on their capability to store information and the rate at which they learn it.

- The analysis of neural entropy on a controllable multivariate Gaussian case gives clear and interpretable results.

- The conclusion on the success of the variance preserving scheme is interesting.

## Weaknesses

- I find the definition of neural entropy a bit confusing. Equation (11) defines the ideal neural entropy as the total entropy, while equation (18) defines the “actual” neural entropy as the subpart of the total entropy that is due to the diffusion model formulation. This shift in definitions creates a confusion on the interpretation of the measure.

---

> ### Author Rebuttal · Authors · 2025-07-30
>
> Thank you for the review and many insightful questions. We address your points below.
>
> Weaknesses:
>
> $S_{\rm tot}$ is the total entropy produced by the forward process. This is also the information that must be supplied to go back to the initial distribution $p_{\rm d}$. In principle $S_{\rm tot}$ can be computed if we knew $\nabla \log p(x,t)$ exactly, using (16) in the paper (we tried to copy the expressions here to make it easier to access, but it does not render). A perfect entropy matching model, with access to infinite data and infinite training time, would make the LHS in (17) zero, in which case $e_\theta = \nabla \log p - \nabla \log p_{\rm eq}^{(t)}$ exactly and the information stored in the NN is the ideal neural entropy $S_{\rm tot}$. But this never happens in practice. Instead $e_\theta$ only approximates this ideal (see Fig. 10). The actual neural entropy $S_{\rm NN}$ is calculated with $e_\theta$. We only introduce the ideal $\hat{S}_{\rm NN}$ for completeness.
>
> Questions:
>
> 1. Reviewer KtyK also raised the question about network size; please refer to our response there for a detailed discussion.  Re: batch size and number of epochs, yes we did experiment with these hyper parameters in the Gaussian mixture + MLP setting. The results are discussed in detail in another paper of ours, which has been cited in *Neural Entropy*, but we refrain from citing it here to stay in compliance with the double-blind reviewing policy. The short summary is that if the number of samples $N$ is too sparse relative to the ambient dimensions, the model overfits and we see the neural entropy rising (see response to KtyK). This also happens if the model trains for a long time. The results were stable over a range of batch sizes. We never experimented with the optimization algorithms. As for the question about resampling Gaussians: we used the same samples in every epoch, with shuffling. This was done to stay as close as possible to the image model experiments.
>
> 2. We did not run experiments on ImageNet due to limited computational resources, but we agree it would be an interesting direction. We hope to run experiments on bigger datasets soon.
>
> 3. See second point in the rebuttal to KtyK. In short, the inductive biases of the neural network impart a certain 'stiffness' to the increase in neural entropy, as shown in Fig. 1 and 3 of the paper. Therefore (image) diffusion models naturally resist memorization, since it would take a very long time to absorb enough neural entropy to reverse diffuse back to Dirac delta functions at the training samples.
>
> Suggestions:
> Thanks for catching that, it will be corrected in future revisions.

---

> > ### Comment · Reviewer_nu8R · 2025-08-06
> >
> > The response given by the authors, complemented by the rebuttal to reviewer KtyK, are convincing enough.

---

### Official Review · Reviewer_KtyK · 2025-07-03

**Clarity:** 3
**Significance:** 3
**Originality:** 3
**Rating:** 5
**Confidence:** 4

**Summary:**

This paper explores the interplay between deep learning and information theory through the lens of diffusion models. It proposes a novel metric, neural entropy, to quantify the amount of information retained within the neural network during training. This measure captures aspects of both the underlying data distribution and the dynamics of the diffusion process. Empirical evaluations on image diffusion tasks demonstrate that these models achieve remarkable efficiency in compressing structured data.

**Questions:**

I would appreciate a detailed and thoughtful response to the concerns raised in the Weaknesses section. If these concerns are adequately addressed, I would be willing to reconsider my evaluation and potentially adjust the score in a more positive direction.

**Ethical Concerns:**

["NO or VERY MINOR ethics concerns only"]

**Final Justification:**

The authors have provided a clear and satisfactory response to several of the concerns I raised in my initial review. I appreciate their effort in addressing these issues, and as a result, I am updating my score accordingly.

**Limitations:**

Yes

**Quality:**

3

**Strengths And Weaknesses:**

Strengths

1. The paper is well-motivated and makes a meaningful contribution to the theoretical understanding of diffusion model training.
2. It is well-written and presents extensive theoretical analysis, supported by empirical results that validate the proposed theory.

Weaknesses

1. The relationship between network size and neural entropy requires further consideration. The paper would benefit from additional theoretical analysis and experiments addressing this relationship. Referring to related studies, such as [A], could strengthen this aspect.
2. Including a discussion on generalization could strengthen the impact of this study. In particular, it would be valuable to investigate whether the model tends to overfit or generalize when training proceeds with new data after entropy saturation. Such an analysis would provide deeper insight into the relationship between entropy dynamics and generalization behavior.

[A] Morris, John X., et al. "How much do language models memorize?." arXiv preprint arXiv:2505.24832 (2025).

---

> ### Author Rebuttal · Authors · 2025-07-30
>
> Thank you for the review and suggestions. We address your concerns below.
>
> We preface our comments by noting that we are actively working to understand the relation between neural entropy, network size, and generalization. Therefore, the fact that you and another reviewer have independently raised these questions is encouraging. Our goal with the present paper was to introduce the concept of neural entropy and substantiate it with theory and experiments. To that end we have only included findings that are clear and unambiguous. While we are eager to share results related to network size and generalization, this research is still a work in progress, but we share some preliminary insights below.
>
> 1. Network size: We conducted some early experiments with the Gaussian mixture + MLP setup where we added/removed layers to the MLP to study the effect on neural entropy., keeping $p_{\rm d}$ and the forward process fixed. That is, the diffusion process produced the same $S_{\rm tot}$ in all these experiments, which is also the information the network needs to store to reverse the process. While it is true that a very small MLP fails in absorbing the entirety of $S_{\rm tot}$, we find that beyond a certain point adding more layers has little benefit. That is, there is some network size at which the NN captures most of $S_{\rm tot}$, and making the network bigger than this has diminishing returns in the quality of reconstruction, as measured by ${\rm KL}(p_{\rm d} || p_\theta)$.
>
>     In image models, this question becomes more nuanced due to the interplay between network size and inductive bias. For example a larger U-net where the inductive biases are weakened, say by using a larger stride in its convolutional layers, may perform worse than a smaller U-net where the stride is set to 1. We are currently investigating this tradeoff in greater depth.
>
> 2. Generalization: Diffusion models tend to overfit/memorize when training on sparse datasets and/or over long training runs [1]. Apparently the strong inductive biases in image diffusion models safeguard against memorization to a certain degree, and much more so than in the Gaussian mixture/MLP case (see paragraph to the left of Fig. 4 in *Neural Entropy*). This is another way to understand Fig. 1 and 16 from our paper. Let us explain.
>
>     Memorization happens when the model learns a reverse drift that drives the initial distribution back to set of Dirac delta functions at the training data. But concentrating the probability mass into such small regions requires that the network supply a far greater amount of information, since information negates entropy and 'thinner' distributions have lower entropy. Or, in terms of Schrodinger's Gedankenexperiment, it is much more improbable for a homogenous distribution to automatically fluctuate into a configuration with very dense slivers of probability, than to a distribution where the probability is more evenly distributed. This is to say that the neural entropy would have to be much higher for the model to memorize. But, as we see in Fig 3. for example, the growth in neural entropy plateaus at later epochs of training, which means the model has to train for much longer to reach a memorized state. So memorization is harder for diffusion models because they resist overfitting due to the information cost. While this does not fully explain generalization, for example why the generated images are sensible, it gives us a promising new tool to approach the problem. We plan to explore this direction in a follow-up work.
>
> We appreciate your suggestion to cite [A] (Morris et al., 2025). While their study focuses on language models, their insights on memorization behavior are indeed relevant.
>
> [1] Gu et. al. "On Memorization in Diffusion Models" arXiv:2310.02664v2 (2025)

---

> > ### Comment · Reviewer_KtyK · 2025-08-08
> >
> > The authors have provided a clear and satisfactory response to several of the concerns I raised in my initial review. I appreciate their effort in addressing these issues, and as a result, I am updating my score accordingly.

---

### Official Review · Reviewer_VsRX · 2025-07-03

**Clarity:** 1
**Significance:** 4
**Originality:** 4
**Rating:** 4
**Confidence:** 1

**Summary:**

This paper aims to measure the information that a trained diffusion model stores about its training data. The main result of the paper is that this information is proportional to $\log n$ where $n$ is the training dataset size, rather than $n$, as would be the case for a memorizing network sampling from the empirical training data distribution.

**Questions:**

What is $S_{\mathrm{NN}}$ measuring? Why would it be the correct measure of information stored in the network? Is it still valid for an untrained network? It seems it can be made arbitrary large for arbitrary networks. What do the equations look like in the variance exploding case ($b_+ = 0$), do the authors still think of them as valid in that setting? Why are the equations not invariant to time reparameterization? Simply rescaling time should not fundamentally change the information needed to store the training distribution.

In my current state of understanding of the paper, I cannot recommend acceptance, but I am willing to improve my score if the authors convincingly demonstrate the well-foundedness of their entropy measure.

**Ethical Concerns:**

["NO or VERY MINOR ethics concerns only"]

**Final Justification:**

This paper presents intriguing results, and even if the quantities it measures are somewhat flawed in my opinion, they perhaps could be refined in future work.

**Limitations:**

yes

**Paper Formatting Concerns:**

It seems the NeurIPS paper template was used in "preprint" mode rather than "submission", leading to the removal of line numbers (which makes writing the review a bit more difficult).

**Quality:**

1

**Strengths And Weaknesses:**

This paper tackles a fundamental question in diffusion generative modeling, and its results are very intriguing. However, in its current state, I find it difficult to understand, especially as a non-physicist.

Some of the physics jargon is difficult to understand for non-physicists: e.g., top of page 2, "the entropy of the phase space density over the neural network’s internal microstates" does not make any sense to me. Referencing "the total entropy produced by a diffusion" without motivating its expression or the Maxwell demon without briefly introducing it is also not appropriate given the audience of NeurIPS. I also don't understand what it means for the marginal distribution of a particle evolving under an SDE to itself fluctuates: by definition the distribution is deterministic, and evolves under an ODE (Fokker-Planck). It seems that what is meant is the empirical distribution of the $M$ walkers, but in that case it cannot be equal to $p_{\mathrm{d}}$ nor $p_{\mathrm{eq}}$.

I end up quite lost in the notation, especially given that some objects are never explicited (e.g., the kernel $h_\star$: does it admit a closed-form equation?) or redefined multiple times. For instance, during the first reading I got confused between $S_{\mathrm{tot}}$, $\hat S_{\mathrm{NN}}$, and $S_{\mathrm{NN}}$, or $s_\theta$ and $e_\theta$, or $p_0$, $p_{\mathrm{eq}}$, and $p_{b_+}$. These quantities are often introduced early in the text, not used for a while, then mentioned later without reminding the reader what they stand for.

The main issue is that after reading the main text and the appendix twice, and spending some time thinking about it, I am not confident at all about what is measured in Figure 1, the central result of the paper. More specifically, I don't see how or why the proposed definition of $S_{\mathrm{tot}}$, and even less so $S_{\mathrm{NN}}$, should coincide with the "information load" on the network. First, in the so-called "variance-exploding" SDE (where $b_+ = 0$), $S_{\mathrm{tot}}$ is simply equal to $H(p_{\mathrm{eq}})$ - $H(p_{\mathrm{d}})$. This does not seem to me to be a reasonable estimate of information "needed to reverse the diffusion process" in that case. Second, after several pages of careful motivation and explanation of $S_{\mathrm{tot}}$, $S_{\mathrm{NN}}$ is introduced instead without explanation. As a result, I have no intuition what it measures.

Finally, the following reference might be relevant to the authors, as it derives several information-theoretic relations in diffusions that are very related to those in this paper:
> Kong, X., Brekelmans, R., & Steeg, G. V. (2023). Information-theoretic diffusion. arXiv preprint arXiv:2302.03792.

To summarize:
- Strengths: the paper studies an important topic, and the claimed results are very significant (though I cannot evaluate them fully due to my limited understanding of the paper)
- Weaknesses: the style of writing is hard to follow, and the final definition of $S_{\mathrm{NN}}$, central to the paper, is not motivated enough (why should it measure the amount of information about the training set stored by the diffusion model?)

---

> ### Author Rebuttal · Authors · 2025-07-30
>
> Thank you for the review, and for engaging with the paper despite the fact that it was outside your area(s) of expertise. We value your feedback as non-physicist, and appreciate your transparency.
>
> We would like to make the paper accessible to a broad audience while remaining within the constraints of the page limit. For this reason, we focused the main text on presenting our novel contributions, rather than review established concepts in physics and information theory. In the appendix we discuss these ideas more gradually and in greater detail, with extensive references that fill in the pre-requisites. Even when we allude to specific physics concepts like phase space densities and Maxwell's demon, we have written it in such a way that skimming over those portions of the paper does not diminish our core message. We would also like to gently push back against the characterization of some of the wording as jargon: this is standard terminology in statistical physics, and is intended to communicate the ideas efficiently rather than to feign sophistication.
>
> With respect to the distribution itself fluctuating, an equation like $\partial_t p(x,t) = \frac{\sigma^2}{2} \nabla^2 p(x,t)$ gives the *dominant* evolution of the distribution. What we hoped to illustrate with the example of the random walkers on the lattice is that finite samples deviate from this average. When we say that the probability of a random walker to jump left (or right) is  $q_L$ or $q_R$, it does not mean that exactly $q_L$ fraction of all the walkers at a site will jump to the left in every single time step. Here is another example: if you take a perfectly fair coin and toss in 1000 times you don't get exactly 500 heads/tails; there will be some fluctuations. In fact there is a small but non-zero chance of getting all heads, $p(\text{all H})=2^{-1000}$. In the same way, the random walkers have a small probability of deviating away from the evolution governed by $q_L$ and $q_R$ at each time, culminating in a vastly different outcome that what one might expect from the dominant evolution. The probability of this happening is very small of course, which is what leads to the form $\mathcal{P}[p_{\rm d}] \approx e^{-M S_{\rm tot}}$. This is an example of the *Large Deviation Principle* [1].
>
> Regarding the notation, we are sorry to hear that the symbols were difficult to follow. The meaning of each of these symbols is explained as they are introduced. For example, we state explicitly under (2) that "$h(x_T|x_0)$ is a kernel that transports $p_{\rm eq}(x_0)$ to $p_{\rm d}(x_T)$" and that $h_\star$ is the optimal kernel than minimizes the KL to $g$. Furthermore, we give a more elaborate discussion of these objects in App. A.2 where the lattice random walker problem is explored in greater detail. The same is true for the other symbols in the list. There is also a paragraph summarizing these symbols at the beginning on App. B, page 18.
>
> The concept of neural entropy defined here requires a confining drift term in the forward diffusion process, because those are the only processes which have an equilibrium state, $p_{\rm eq}$, that results from a trade-off between the confining effect of the drift and the dispersive effect of the noise. As explained above, $S_{\rm tot}$, and therefore the neural entropy, is related to the probability that the system fluctuates away from the equilibrium state. The variance preserving term does not have an equilibrium state, and therefore we cannot define neural entropy in that case. This is also related to the discussion in App. D. The connection between $S_{\rm tot}$ and $S_{\rm NN}$ is explained in the response to the review by nu8R. Briefly, the network must absorb $S_{\rm tot}$ amount of information to reverse diffuse perfectly.  However, due to imperfections in training and finiteness of the training data it can only asymptote to this ideal. Figs. 2 and 10 in the paper show how $S_{\rm NN}$ approaches $S_{\rm tot}$ over the course of training.
>
> Questions:
> We believe we have addressed several of your questions in our comments above. We answer the remaining ones here.
>
> 1. Is neural entropy valid for an untrained network? Answer: No. Neural entropy measures the amount of information stored in the network rather than total network capacity. An analogy would be a hard drive: if you store a 1GB file, the drive contains 1GB of information, regardless of whether its capacity is 10GB or 100GB.
>
> 2. To understand the effect of time rescaling, consider the VP process ${\rm d} Y = -\frac{\beta(s)}{2} Y {\rm d} s + \sqrt{\beta(s)} {\rm d} B$, which is applied for a time $T$. We can rescale this by introducing a new time variable ${\rm d} \tau = \beta(s) {\rm d} s$, which converts the SDE to ${\rm d} Y = -\frac{1}{2} Y {\rm d} \tau + {\rm d} B$. However, the latter has to be applied for a time $T' = \int_{0}^{T} \beta(s) {\rm d} s$ to reach the same state. It is then straightforward to see, from the expression for neural entropy and (25) in the paper, that both SDEs produce the *same* $S_{\rm tot}$. So entropy production is indeed invariant to time rescaling, as you correctly point out.
>
> We hope this response clarifies the key concepts and addresses your concerns. Thank you for acknowledging the significance of our work, and being forthright with your self-assessment.
>
> [1] H. Touchette, "The large deviation approach to statistical mechanics", arXiv:0804.0327v2 (2009)

---

> > ### Comment · Reviewer_VsRX · 2025-08-04
> >
> > I thank the authors for their reply.
> >
> > My current understanding of the paper is the following. In the case of time-homogeneous dynamics that relaxes towards an equilibrium, $S_{\rm tot}$ quantifies the information needed to communicate one sample of $p_{\rm d}$ to someone that uses the "prior" defined by the diffusion process, and is roughly $D_{\rm KL}(p_{\rm d}||p_{\rm eq})$ (it seems that this is not an equality because of subtleties that escape me). This is well established. The paper then proposes two extensions of this entropy: (1) a generalization to non-equilibrium processes, and (2) a generalization to imperfectly trained networks. I am still not entirely convinced of the well-foundedness of these two extensions, although the empirical results demonstrate that it is well-behaved. In particular, I am slightly suspicious still that the framework doesn't apply to the variance-exploding case. Intuitively, going from variance-preserving to variance-exploding (or any other linear-drift constant-diffusion SDE) is a simple change of variable, and it seems to me that it should not meaningfully change the answer. For instance, I would expect the information load on the network to learn $e_\theta$ or $e_\theta + \nabla \log p_{\rm eq}^{(t)}$ to be the same (or at least that they have the same scaling in the sample size $n$).
> >
> > The reason for my question on invariance to time reparameterization is the speed limit equation (22). How can it make sense if $S_{\rm tot}$ is independent of $T$?
> >
> > I now understand the large deviation meaning of equation (2). It was confusing for me as the empirical distribution of $M$ walkers is a sum of $M$ Dirac delta functions, and thus cannot be equal to $p_{\rm d}$. Simultaneously, fluctuations are suppressed in the $M\to\infty$ limit. So making this statement precise is quite subtle, and the text could benefit from some clarifications there.
> >
> > Regarding untrained networks: I understand that neural entropy does not measure absolute network capacity, but the "used" capacity (by a loose analogy with information theory, that could be defined as the "mutual information" between the network weights and the training data, rather than the "entropy" of the weights). So we expect that $S_{\rm NN} = 0$ for an untrained network. But it seems to me that this value is just a function of the scale of the network initialization, and could thus be much greater than the neural entropy at the end of training. In that case, I suppose it would indicate that a random network at initialization encodes a non-trivial distribution that does not admit a compact description. In that sense, neural entropy measures more the complexity of the generative model than its informativeness about the training data?

---

> ### Author Response · Authors · 2025-08-04
> **Further clarification**
>
> Thank you for the questions.
>
> 1. $S_{\rm tot}$ is the information required to reconstruct the *entirety* of $p_{\rm d}$ from the prior $p_{\rm eq}$. We can extract from this the information needed to produce just one sample, let's call it $s_{\rm tot}(x)$, in which case $S_{\rm tot} = E_x[s_{\rm tot}(x)]$. Similar slicing of entropies can be found in [1]. In Sec. 2 of the paper, we normalize the log of the exponent by $M$ to get an $M$-independent quantity. Your high level understanding of the two extensions is correct. The first one, about generalizing to the non-homogenous in time case is explained in great detail in the appendix (we are supposing you meant non-homogenous rather than non-equilibrium, because the time-homogenous evolution is also non-equilibrium). The reason why $S_{\rm tot} \neq {\rm KL}(p_{\rm d} || p_{\rm eq})$ is also given there; in a finite time $T$ the system does not fully equilibrate, it reaches a $p_0$ that is close to $p_{\rm eq}$ but not quite $p_{\rm eq}$. In other words, some information is still left behind at time $T$, therefore the KL between these two have to be subtracted from ${\rm KL}(p_{\rm d} || p_{\rm eq})$. This is the meaning of (15). The second extension applies when the approximation $e_\theta \approx \nabla \log p - \nabla \log p^{(t)}_{\rm eq}$ holds. See discussion in the rebuttal to reviewer nu8R, under 'Weaknesses'.
>
>     To address the VP/VE question, let us consider explicitly the transformation of a linear-drift SDE to one without a drift term. We start with the SDE ${\rm d} X_t = -\gamma X_t {\rm d}t + {\rm d}B_t$, where $\gamma$ is a positive constant. Setting $Y_t = e^{\gamma t} X_t$ would give us the SDE ${\rm d} Y_t = e^{\gamma t} {\rm d} B_t$, which is unbounded Brownian motion where the noise grows exponentially in strength over time. This is because the change of variables we had to do to remove the drift term amplified the randomness in $X$. Thus the entropy production in $Y$ is *not* the same as that of $X$.
>
> 2. $S_{\rm tot}$ is *not* independent of $T$. This should be apparent from (15), where $T$-dependence enters as the upper limit of the integral. $S_{\rm tot}$ also depends on $p_{\rm d}$ and the forward diffusion coefficients. This is why we did not explicitly indicate its $T$-dependence early in the text. However we do make the time dependence explicit from (23) onwards, wherever it is necessary.
>
> 3. In the lattice case, the distribution $p_{\rm d}$ is a sum of Kronecker delta function at the lattice sites, not Dirac deltas. But we see what you are trying to say in the context of the continuum case.
>
> 4. Your intuition about the mutual information between neural weights and the training data is correct. What you refer to as *"entropy" of the weights* is what we meant by entropy over the phase space density of microstates of the network earlier. We have difficulty understanding your statements from "But it seems to me that this value is just a function of the scale of the network initialization..." onwards.
>
> [1] U. Seifert, Entropy Production along a Stochastic Trajectory and an Integral Fluctuation Theorem, Phys. Rev. Lett. 95, 040602.

---

> > ### Comment · Reviewer_VsRX · 2025-08-04
> >
> > Thank you. I have no further questions. I think this paper presents intriguing results, and even if the quantities it measures are somewhat flawed in my opinion, they perhaps could be refined in future work. I hope that this discussion help the authors increase the clarity of the paper, especially with a non-physicist audience in mind. I have increased my score to a 4.

---

> > > ### Author Response · Authors · 2025-08-05
> > > **Thank you**
> > >
> > > Thank you for the discussion, and for updating your score. We appreciate your perspective.

---

### Note · Authors · 2025-08-14

We are grateful to the reviewers for their thoughtful feedback. We are pleased that all reviewers found the paper intriguing and that at least three of the four increased their scores following the exchange.

We note that most critical comments from reviewers were not about flaws or errors, but rather about what more could be done: more datasets, more diagnostics, larger models, deeper empirical analysis. We take these as positive signals; a clear indication that reviewers saw the potential scope and applicability of neural entropy beyond what we could fit in the current submission. These directions are indeed part of our ongoing and future work.

Reviewer VsRX initially expressed skepticism, citing unfamiliarity with the physics foundations. We gave detailed technical rebuttals addressing each of their concerns. Although they maintained some reservations, they acknowledged the value of the work and raised their score from 2 to 4. We appreciate their willingness to revise their opinion and hope our response improved clarity for readers beyond the physics community.

We appreciate the opportunity to present this work and to engage in open discussion. The nature of the questions posed, and the subsequent score updates, reinforce our belief that this work makes a timely and substantive contribution. We feel encouraged to build on it in follow-up research and thank you for your consideration during the decision process.

---

### Decision · Program_Chairs · 2025-09-17

**Decision:**

Accept (spotlight)

**Comment:**

The reviewers agreed that the paper tackles a fundamental question in diffusion generative modeling, offering intriguing results. This is also an original contribution that connects diffusion models with information theory, presenting extensive theoretical analyses which are complemented by empirical results, giving several important insights on neural networks, namely on their capability to store information and the rate at which they learn it, whereas the analysis of neural entropy on a controllable multivariate Gaussian case gives clear and interpretable results. The conclusion on the success of the variance preserving scheme was found interesting. There were favourable comments on this manuscript, including clarity and readability, intuitive explanations for the amount of information stored into a network used to reverse the dynamics of diffusion processes, relation between information storage and the structure of the data, the number of training samples, and the particular form of the stochastic process used is important and could be used to better engineer diffusion-based generative models, and introducing the entropy-matching objective, which is easy to implement and optimize, similarly to flow-matching objectives, although care needs to be taken to select appropriate measures for numerical stability.

There was clear support for the recommendation to accept this work. In the rebuttals the authors diligently committed to implement some clarifications and improvements on the manuscript, to address concerns on the specialised jargon and the need to improve clarity on some definitions and motivations, as well as practical implications. I strongly encourage the authors to integrate the feedback received into the revised version of their paper.